# Long-term live imaging of the *Drosophila* adult midgut reveals real-time dynamics of division, differentiation and loss

**Judy Lisette Martin[1]**, **Erin Nicole Sanders[1,2]**, **Paola Moreno-Roman[1,3]**, **Leslie Ann Jaramillo Koyama[1,2]**, **Shruthi Balachandra[1]**, **XinXin Du[1]**, **Lucy Erin O'Brien[1]\***

[1]Department of Molecular and Cellular Physiology, Stanford University School of Medicine, Stanford, United States; [2]Department of Developmental Biology, Stanford University School of Medicine, Stanford, United States; [3]Department of Biology, Stanford University, Stanford, United States

**\*For correspondence:**
lucye@stanford.edu

**Competing interests:** The authors declare that no competing interests exist.

**Abstract** Organ renewal is governed by the dynamics of cell division, differentiation and loss. To study these dynamics in real time, we present a platform for extended live imaging of the adult *Drosophila* midgut, a premier genetic model for stem-cell-based organs. A window cut into a living animal allows the midgut to be imaged while intact and physiologically functioning. This approach prolongs imaging sessions to 12–16 hr and yields movies that document cell and tissue dynamics at vivid spatiotemporal resolution. By applying a pipeline for movie processing and analysis, we uncover new and intriguing cell behaviors: that mitotic stem cells dynamically re-orient, that daughter cells use slow kinetics of Notch activation to reach a fate-specifying threshold, and that enterocytes extrude via ratcheted constriction of a junctional ring. By enabling real-time study of midgut phenomena that were previously inaccessible, our platform opens a new realm for dynamic understanding of adult organ renewal.

DOI: https://doi.org/10.7554/eLife.36248.001

## Introduction

Stem-cell-based organs rely upon the coordinated control of cell division, differentiation and loss to maintain tissue homeostasis. Studies of the *Drosophila* adult midgut (*Figure 1A*) have elucidated conserved processes and pathways that control these events during healthy turnover and cause their dysfunction during aging and in cancer. These contributions, which include descriptions of the mechanisms of multipotency and asymmetric-symmetric fates, endocrine and immune regulation, and injury and stress responses, span the range of adult stem cell biology (*Biteau et al., 2008*; *Buchon et al., 2009*; *Deng et al., 2015*; *Guo and Ohlstein, 2015*; *Hudry et al., 2016*; *Jiang et al., 2009*; *O'Brien et al., 2011*; *Ohlstein and Spradling, 2007*; *Siudeja et al., 2015*).

Nevertheless, investigation of midgut cell dynamics has been constrained by the lack of a viable platform for extended live imaging. At present, fixed midguts provide static snapshots of cells and tissues but do not allow dynamic behaviors to be observed over time. Meanwhile, cultured midguts have been imaged ex vivo for 60–90 min—a time window long enough to allow studies of faster events such as calcium oscillations, cell divisions and acute toxicity responses (*Antonello et al., 2015*; *Deng et al., 2015*; *Lee et al., 2016*; *Montagne and Gonzalez-Gaitan, 2014*; *Scopelliti et al., 2014*), but too short for observations of slower events such as differentiation and apoptosis. Indeed, the power of extended live imaging is demonstrated by studies of numerous other stem-cell-based organs, including *Drosophila* ovary and testis (*Fichelson et al., 2009*; *Lenhart and DiNardo, 2015*; *Morris and Spradling, 2011*) and mouse epidermis, testis, muscle and intestine (*Bruens et al.,*

*2017*; *Gurevich et al., 2016*; *Hara et al., 2014*; *Ritsma et al., 2014*; *Rompolas et al., 2012Rompolas et al., 2016*; *Webster et al., 2016*). For the midgut, long-term live imaging would synergize with the organ's existing genetic tractability and well-characterized cell lineages to open exciting investigative possibilities.

To enable such studies, we present a simple platform that substantially extends imaging times by keeping the midgut within a living animal. The live animal is secured in a petri dish, and the midgut is visualized through a window cut into the dorsal cuticle. The organ's structural integrity stays largely intact, which allows routine acquisition of movies of ~12–16 hr in duration. Furthermore, digestive function is preserved; the animals ingest food, undergo peristalsis, and defecate even while being imaged. As a result, these long-term movies vividly capture midgut cell dynamics in a near-native physiological context.

To allow mining of the data in these movies, we also present a systematic approach for image processing and segmentation and for the spatiotemporal analysis of single cells and whole populations. These proof-of-principle analyses both corroborate prior fixed-gut observations and reveal intriguing, dynamic behaviors that relate to cell division, differentiation and loss. (1) For division, we find that mitotic stem cells frequently re-orient—sometimes repeatedly—but can be 'anchored' in place by two immature enteroblast cells. (2) For differentiation, we analyze the kinetics of Notch activation, which reveal the transition from a stem-like to a terminal cell state, and we find that, contrary to expectation, real-time activation does not correlate with contact between Notch- and Delta-expressing siblings. (3) For cell loss, we perform morphometric analysis of enterocyte cell extrusion over time and find that extrusion occurs via ratcheted constriction of a basal junctional ring. These analyses demonstrate the power of examining midgut cell dynamics in a near-native context over multi-hour timescales. By allowing real-time observation of cellular events that were previously inaccessible, our platform holds promise to advance our understanding of the fundamental cell behaviors that underlie organ renewal.

## Results and discussion

### An apparatus for midgut imaging within live *Drosophila* adults

We designed a 'fly mount' for imaging the midgut in live *Drosophila* adults (*Figure 1B*). Our mount, similar to an apparatus for imaging adult *Drosophila* brains (*Seelig et al., 2010*), is assembled from inexpensive, common materials and can be configured for upright, inverted or light-sheet microscopes (*Figure 1—figure supplement 1A–D*). In the mount, a live animal is stabilized by affixing its abdomen in a cutout within a petri dish (for upright or inverted microscopes) or a syringe barrel (for light-sheet microscopes) (*Figure 1—figure supplement 1A–D*, *Figure 1—figure supplement 2*, *Video 1*). The midgut's R4a-b (P1-2) region (*Buchon et al., 2013a*; *Marianes and Spradling, 2013*) is exposed through a window that is cut in the dorsal cuticle (*Figure 1B–C*, *Video 1*). This arrangement leaves the midgut-associated trachea and neurons largely intact (*Video 2*). Steps to assemble the fly mount and prepare the midgut are illustrated in a detailed tutorial (*Video 1*).

Three design features prolong animal viability. First, the animal is provided liquid nutrition through a feeder tube and allowed to 'breathe' through unoccluded spiracles (*Figure 1B*, *Figure 1—figure supplement 1A*). Second, the exposed organ is stabilized by an agarose bed and bathed in media (*Figure 1C*). Third, the animal is kept hydrated in a humidity box (*Figure 1—figure supplement 1B*). Throughout imaging, animals continue to ingest food, undergo peristalsis, and defecate, which suggests that midguts remain in a state that approaches native physiology.

A crucial element is the use of a 20x, high-NA dipping objective, which captures z-stacks that are both wide field (100–300 cells) and high resolution (~1 μm) (*Video 3*). Time intervals between z-stacks ranged from 5 to 15 min. At room temperature, 72% of animals were alive and responsive after 12–16 hr of continuous imaging (N = 18 animals; median imaging duration, 14.6 hr) (*Figure 1—source data 1*, *Video 4*). Nearly all cells remained viable, as revealed by the cell death marker Sytox Green (93–98% viability; *Figure 1—figure supplement 3*, *Video 5*). At elevated temperatures (≥29°C), however, the midgut was prone to rupture, so temperature-controlled gene expression by GAL80[ts] or heat-shock induction proved impracticable. Progesterone-induced GeneSwitch drivers (*Mathur et al., 2010*) could be a feasible alternative.

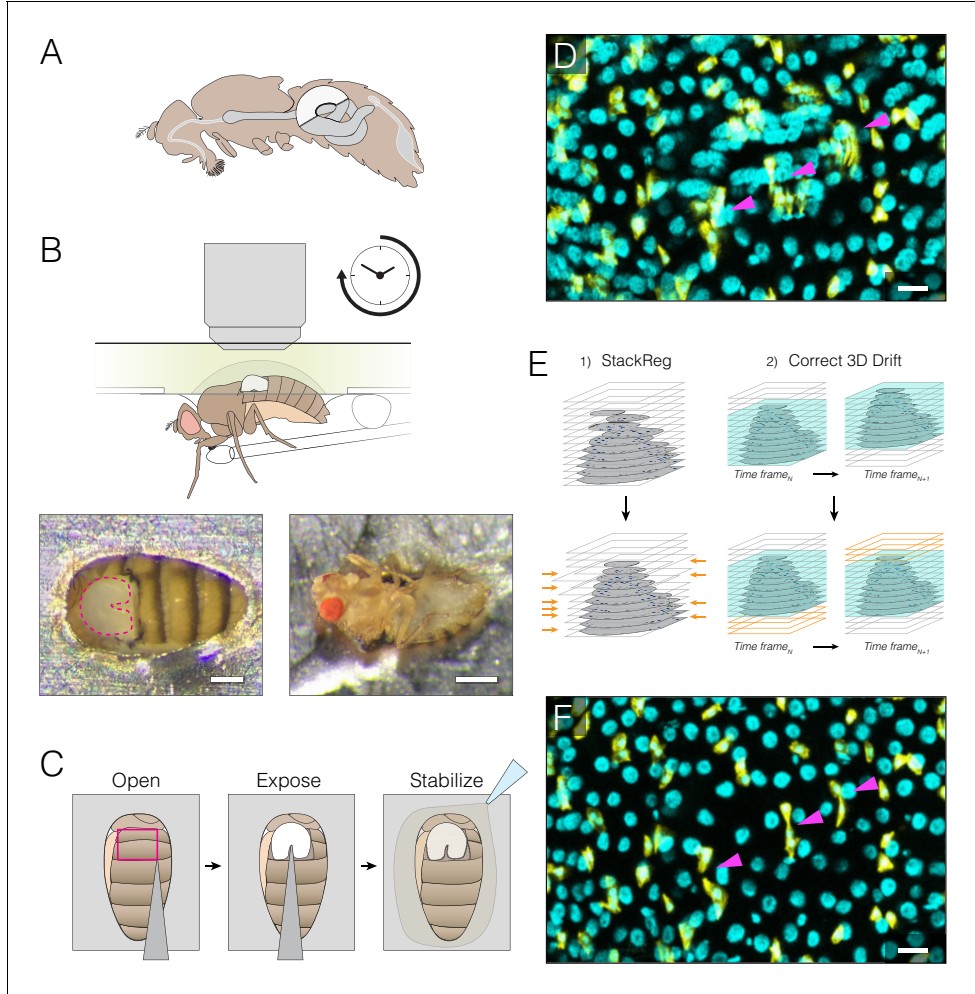

**Figure 1.** Extended imaging of the midgut in live *Drosophila* adults. (**A**) Adult female midgut in situ, sagittal view. The white highlighted area indicates region R4a-b, also known as P1-2, (***Buchon et al., 2013a***; ***Marianes and Spradling, 2013***)) of the midgut that will be exposed for imaging. (**B–C**) The midgut is accessed through a small cuticular window cut in the back of a live animal. (**B**) (Top) Schematic of the imaging apparatus. The animal is affixed to a modified petri dish 'mount'. The chamber of the mount contains media. The underside of the mount supports a feeder tube. See and ***Fig. 1-fig. supplement 2***. (Bottom) Dorsal (left) and ventral (right) views of an animal in the mount. In the left panel, the exposed midgut is outlined by the magenta dotted line. Scale bars: 0.25 mm (left), 0.5 mm (right). See ***Video 4***. (**C**), Steps in preparing the midgut for imaging. See ***Video 1*** tutorial. (**D–F**) Registration macros are applied post-acquisition to correct the blurring caused by tissue movements. (**D**), Before registration, blurring and duplications (arrowheads) are evident. This panel is a raw z-series projection of one movie time point. (**E**), During registration, two ImageJ plugins are applied in series. (1) 'StackReg' corrects for tissue movement during z-stack acquisition at a single time point. (2) 'Correct 3D Drift' corrects for global volume movements over multiple time points. (**F**), After registration, blurring and duplications are negligible. Cyan, all nuclei (*ubi-his2av::mRFP*); yellow, stem cells and enteroblasts (*esg >LifeactGFP*). Scale bars, 20 µm. See ***Video 6***.

DOI: https://doi.org/10.7554/eLife.36248.002

The following source data and figure supplements are available for figure 1:

**Source data 1.** Durations, genotypes, animal ages, and animal viability for movies analyzed in this study.
DOI: https://doi.org/10.7554/eLife.36248.006
**Figure supplement 1.** Mounts for upright, inverted and light-sheet microscopes.
DOI: https://doi.org/10.7554/eLife.36248.003
**Figure supplement 2.** Specifications for abdomen cutouts.
DOI: https://doi.org/10.7554/eLife.36248.004
**Figure supplement 3.** Cell viability during extended imaging.
DOI: https://doi.org/10.7554/eLife.36248.005

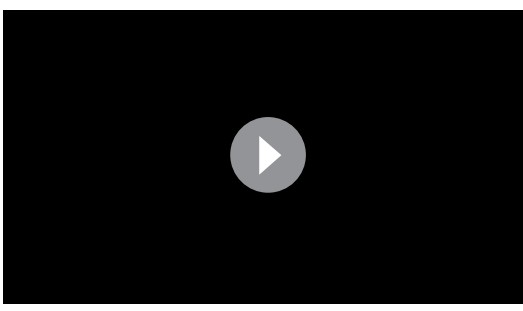

**Video 1.** Narrated, step-by-step tutorial illustrating the preparation of an animal for midgut imaging in the fly mount.
DOI: https://doi.org/10.7554/eLife.36248.007

To minimize interference with native digestion, we used no anesthetics. Hence, ~90% of the raw movies were blurred due to involuntary midgut contractions and voluntary animal movements (*Figure 1D*). In ~30% of these cases, the blurring was too severe for subsequent analysis. In the other ~70%, the blurring could be corrected by the sequential application of two ImageJ macros, StackReg and Correct 3D Drift (*Arganda-Carreras et al., 2006a*; *Parslow et al., 2014*) (*Figure 1E–F*, Source Code File 1, *Video 6*), rendering these movies suitable for single-cell tracking and analysis.

## A systematic approach for comprehensive spatiotemporal tracking of single cells

The study of dynamic cellular events requires that individual cells be identified, tracked and analyzed in space and over time. To facilitate these analyses, we generated a 'fate sensor' line with fluorescent, nuclear-localized markers to allow live identification of the midgut's four major cell types (*esg-GAL4, UAShis::CFP, GBE-Su(H)-GFP:nls; ubi-his::RFP*) (*Figure 2A–B*; *Video 7*). (1) Stem cells are marked by CFP and RFP. Stem cells are responsible for virtually all cell divisions. (2) Enteroblasts are marked by CFP, GFP and RFP. Enteroblasts are Notch-activated stem cell progeny that will mature into enterocytes. (3) Enterocytes are marked by RFP and have polyploid nuclei. Enterocytes are terminally differentiated cells that absorb nutrients and that form the bulk of the epithelium. (4) Enteroendocrine cells are marked by RFP and have small, diploid nuclei. Enteroendocrine cells are terminally differentiated cells that secrete enteric hormones.

To analyze these multichannel, volumetric movies, we developed a semi-automated workflow. ImageJ and Bitplane Imaris are used to separate marked populations digitally, to identify all cells in each population, and to track these cells for the duration of the movie (*Figure 2C–D*). Comprehensive, single-cell tracking enables features such as fluorescence intensity, spatial position and

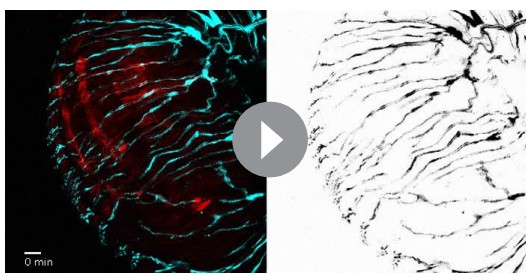

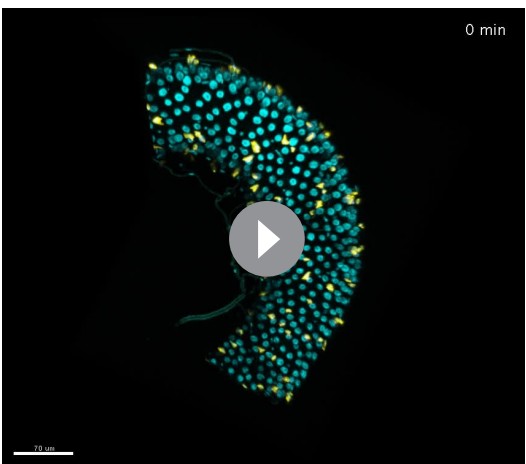

**Video 2.** Movie showing the association of the trachea (cyan) with the midgut tube (red). Smaller tracheal branches encircle the tube and move in concert with peristaltic contractions. A large tracheal branch (upper right) is continuous with smaller branches. The large branch does not move during peristalsis because it is not physically associated with the midgut; instead, it connects the midgut-associated branches to a spiracle (not visible in the movie frame). Left video: cyan pseudocolor, trachea (*breathlessGal4, UAScyt-GFP*); red pseudocolor, microtubules (SiR-tubulin). Right video: inverted gray, *breathlessGal4, UAScyt-GFP*. Each time point is the projection of a confocal z-stack. Scale bar, 20 μm.
DOI: https://doi.org/10.7554/eLife.36248.008

**Video 3.** Volumetric movie of the midgut illustrates the wide-field, high-resolution images that are acquired. Numerous physiological contractions of the midgut are evident. A midgut-associated tracheal branch is visible in the lower left of the video. Scale bar, 70 μm.
DOI: https://doi.org/10.7554/eLife.36248.009

nuclear size to be measured for each individual cell. By multiplying the 100–300 cells in a movie over the hundreds of time points in a 12–16 hr imaging session, we collect tens of thousands of real-time measurements. Unlike prior approaches, which relied on the manual identification and tracking of a few cells, our approach generates single-cell and population-level data in an unbiased manner.

To demonstrate the utility of this imaging platform and workflow, we performed proof-of-principle analyses for three core behaviors of midgut renewal: enterocyte extrusion and loss (*Figure 3A–F*), stem cell division (*Figures 3G–H* and *4*) and enteroblast differentiation (*Figures 5–6*).

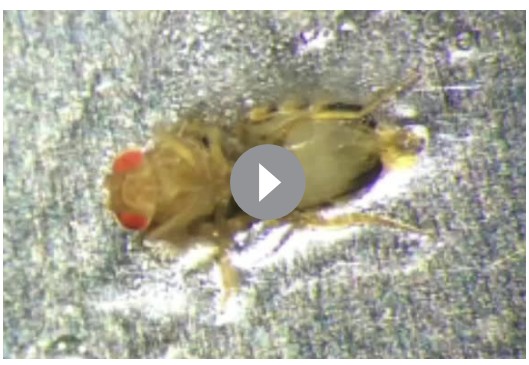

**Video 4.** After 16 hr of continuous imaging, the animal is alive and responsive.
DOI: https://doi.org/10.7554/eLife.36248.010

## Enterocyte extrusion: Spatiotemporal dynamics of ring closure, ring travel, and nuclear travel

Enterocytes in the *Drosophila* midgut, like enterocytes in the mammalian intestine, are lost through apical extrusion (*Buchon et al., 2010*; *Eisenhoffer et al., 2012*; *Harding and Morris, 1977*; *Madara, 1990*; *O'Brien et al., 2011*). During extrusion, a cell is ejected out of the epithelium and into the lumen by the concerted contractions of its neighbors (*Eisenhoffer and Rosenblatt, 2013*). Because this process is seamless, extrusion eliminates apoptotic cells while preserving the epithelial barrier (*Gudipaty and Rosenblatt, 2017*). Apoptotic enterocytes secrete stem cell-activating mitogens (*Liang et al., 2017*), so understanding when and how apoptotic enterocytes are extruded is important for understanding midgut turnover.

In fixed tissues, studies of extrusion have been challenging because extruded cells leave no trace in the epithelium. Although fixed sections can catch extruding cells 'in the act', they do not reveal the dynamics of these transient events.

Our imaging platform enabled us to study extrusions live. Most extrusions were enterocytes, which exited the epithelium either as single cells (18 of 34 total extrusions in six independent movies; *Figure 3A,F*, *Figure 3—figure supplement 1*; *Videos 8* and *9*) or as clusters of 2–5 cells (16 of 34 total extrusions). We also observed one extrusion of an enteroendocrine cell (*Video 10*). Extrusions were distributed comparably across the first and second halves of individual movies. All extrusions were apical.

To gain insight into extrusion dynamics, we performed fine-grained morphometric analysis on three single-enterocyte extrusions (*Figure 3A–E*; *Figure 3—figure supplement 1*). Enterocytes exited the epithelium through constriction of a basal junctional 'ring'. Junctional rings were six-sided and marked by E-cadherin::YFP and myosin::GFP (myosin light chain kinase, or *sqh::GFP*) (*Figure 3A*, *Video 8*). As extrusions progressed, rings closed to a point and eventually vanished (*Figure 3B*, *Video 8*). Meanwhile, neighbor enterocytes drew into rosettes with the extruding cells at their center. Ring closure required ~4–6

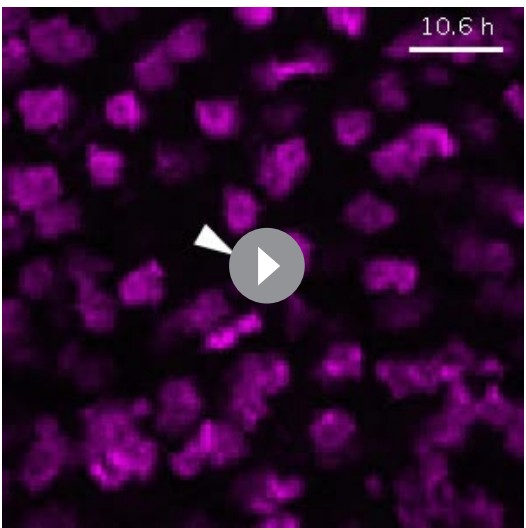

**Video 5.** Cell viability during extended imaging. As cells die, they become marked by the cell death stain Sytox Green, which is continuously present in the imaging media. After 11 hr of imaging, an individual midgut enterocyte changes from Sytox[−] (arrowhead, 10.6 hr), to faintly Sytox[+] (arrowhead, 11.1 hr), to strongly Sytox[+] (arrowhead, 12.0 hr). Nuclei are magenta (*ubi-his2av::mRFP*). Each movie time point is the projection of a confocal z-stack. Scale bar, 20 µm.
DOI: https://doi.org/10.7554/eLife.36248.011

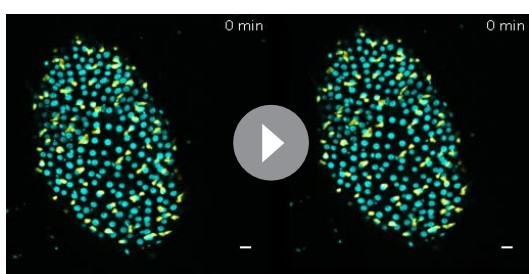

**Video 6.** Movie clip of midgut before (left) and after (right) stack registration. Before registration, blurred cells from tissue movements are evident during timepoints from 20–60 min. After registration, the blurring is negligible. Cyan, all nuclei (*ubi-his2ab:: mRFP*); yellow, stem cells and enteroblasts (*esg >LifeactGFP*). Each time point is the projection of a confocal z-stack. Scale bar, 20 μm.
DOI: https://doi.org/10.7554/eLife.36248.012

hr for completion (*Figure 3—figure supplement 1D*).

Unexpectedly, we found that ring closure was pulsatile and ratcheted. Ratcheted processes are characterized by pulses of constriction that alternate with pulses of stabilization or even relaxation (*Coravos et al., 2017*). All three extrusions exhibited 6–12 of such alternating pulses (*Figure 3C*, *Figure 3—figure supplement 1A–C*). Cumulative times of constriction were similar to cumulative times of stabilization/relaxation. By contrast, rates of constriction generally exceeded rates of relaxation, which drove net closure of the ring over time (*Figure 3—figure supplement 1A–C*).

Ratcheting has not previously been implicated in cell extrusion (*Kuipers et al., 2014*) but is well-known to drive cell deformation in early embryogenesis (*Martin et al., 2009*; *Rauzi et al., 2010*). In embryos, however, pulses are apical, last only 1–2 min, and reshape cells without removing them. By contrast, in enterocyte extrusion, pulses are basal, last 30–60 min, and are associated with delamination.

For an extruding cell to be shed from the epithelium, the junctional ring must not only close but also travel apically toward the lumen. We compared apical travel of the ring to that of the cell nucleus for the extrusion in *Figure 3A*. The ring advanced slowly, with stuttering, apical-and-basal movements that produced net apical progress over 5 hr of ring closure (*Figure 3D*). By contrast, the nucleus shot out of the epithelium in only 15 min (*Figure 3D–E*; *Video 9*, t=150–165 min) and continued to penetrate deeper into the lumen over the next 1.6 hr (*Figure 3D–E*; *Video 9*, t=165–263 min). After reaching maximum depth, the nucleus recoiled and came to rest on the apical epithelium (*Figure 3D–E*; *Video 9*, t=263–443 min). These distinct kinetics suggest that the ring and the nucleus use different mechanisms for apical travel.

Altogether, these analyses provide first morphometric insights on homeostatic cell extrusions in real time. They demonstrate the ability of our platform to reveal novel extrusion behaviors, such as ratcheting, and to enable direct comparison of concurrent subcellular events, such as ring and nuclear travel. Through these abilities, our platform opens the door to a dynamic and quantitative understanding of cell extrusion during organ turnover.

## Stem cell division: Mitotic orientation in real time

Tissue homeostasis requires the replacement of extruded cells by new cells. In the midgut, new cells are generated through stem cell divisions, and terminal daughters are typically post-mitotic. Although time-lapse imaging has unique potential to reveal division behaviors (*Park et al., 2016*), the divisions of midgut cells have been challenging to capture. To date, live divisions have been reported in only one study, which examined pathogen-stimulated midguts ex vivo (*Montagne and Gonzalez-Gaitan, 2014*).

We surveyed our movies of near-native midguts for physiological divisions. Thirty-nine mitoses were identified in 11 independent movies, which had a combined duration of 122 hr. The average mitosis lasted 43 ± 11 min (*Figure 3G–H*; *Figure 3—source data 1*, *Video 11*). Together, these measurements imply a mitotic index of 0.28% (see 'Materials and methods' for calculation), which is less than the 1–2% mitotic index obtained from counts of phospho-histone H3[+] cells in fixed midguts (*Jin et al., 2017*; *Kolahgar et al., 2015*; *Montagne and Gonzalez-Gaitan, 2014*). No divisions were identified in 14 additional movies. Within individual movies, division rates did not trend upward or downward over time. This absence of drift suggests that division rates were reduced not by cumulative imaging stress, but rather by elements that were already present when imaging began or that occurred stochastically. To lessen this inhibitory effect, adjustments to the media formulation would be one attractive approach.

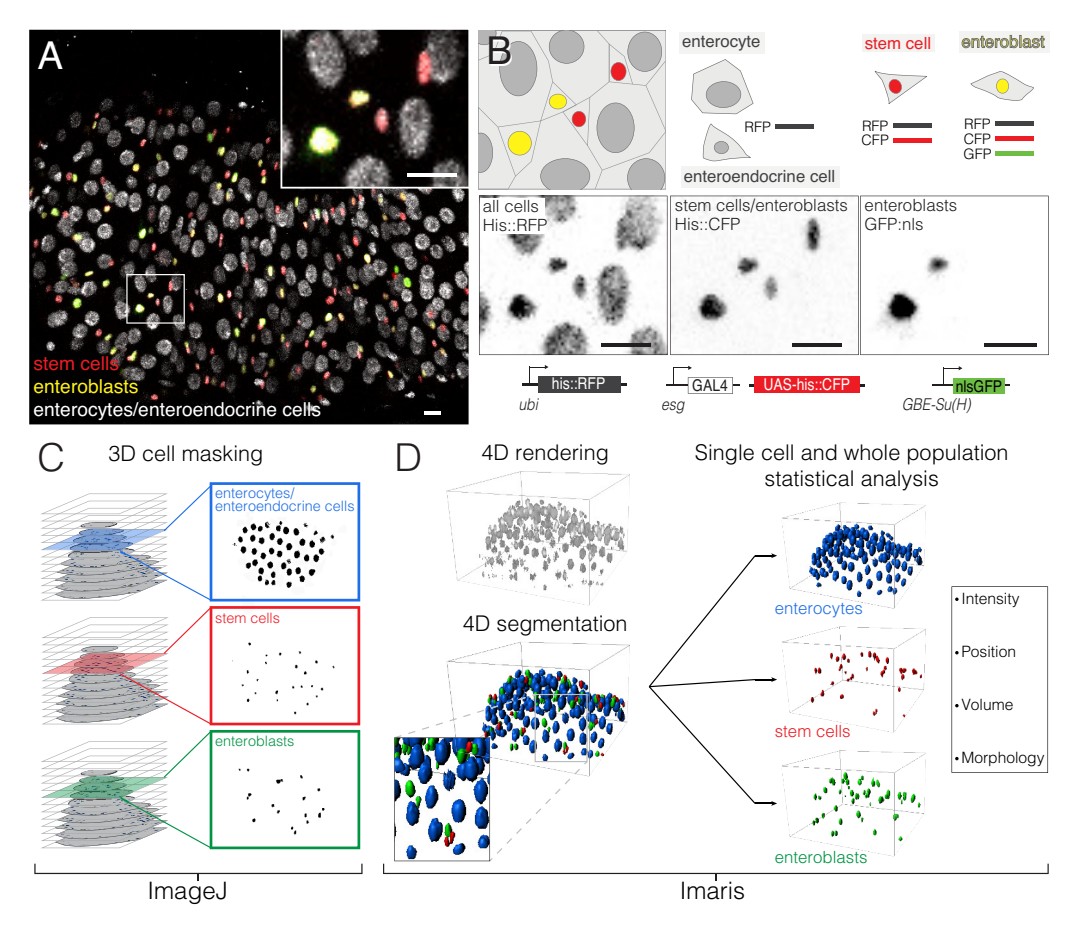

**Figure 2.** Comprehensive, fate-specific tracking and analysis of individual cells. (A–B) 'Fate sensor' midguts enable the live identification of cell types. (A) Stack projection of a single time point from a 10 hr movie (*Video 7*). Nuclei are distinguishable for four midgut cell types: stem cells (red pseudocolor), enteroblasts (yellow-green pseudocolor), enterocytes (gray, polyploid), and enteroendocrine cells (gray, diploid). Inset shows the zoom region depicted in (B). (B) Genetic design of the fate sensor line (*esg >his2b::CFP, GBE-Su(H)-GFP:nls; ubi-his2av::mRFP*). Cell types are distinguished by the combinatorial expression of three fluorescent, nuclear-localized markers: enterocytes/enteroendocrine cells (His2ab::mRFP only), stem cells (His2ab::mRFP, His2b::CFP), and enteroblasts (His2ab::mRFP, His2b::CFP, GFP:nls). All scale bars, 10 μm. (C–D) Workflow to identify, track and analyze cells in volumetric movies. (C) Nuclei from raw, multi-channel z-stacks are digitally separated into stem cell, enteroblast, and enterocyte/entero-endocrine populations using channel masks in ImageJ. (D) The three population sets are rendered in 4D in Imaris. Segmentation is performed on each population to identify individual nuclei. Enteroendocrine nuclei are separated from enterocyte nuclei by a size filter. The positions of individual nuclei are correlated between time points to track single cells over time.

DOI: https://doi.org/10.7554/eLife.36248.013

Each mitosis was exhibited by a unique cell. The vast majority of these were presumably stem cells. However, a recently described cell type, the enteroendocrine precursor cell, accounts for <5% of mitoses (*Chen et al., 2018*). Our movies lacked markers to distinguish stem cells from enteroen-docrine precursors, so the latter may have been responsible for some observed mitoses.

We investigated how mitotic cells dynamically orient in 3D space. In general, epithelial divisions can be considered in two orthogonal frames of reference: horizontal-vertical and longitudinal-circum-ferential. Horizontal-vertical orientation is defined by the epithelial plane (*Figure 4A*) and, in development, serves to determine daughter fates (*Cayouette and Raff, 2003*; *Dong et al., 2012*; *Guo and Ohlstein, 2015*; *Williams et al., 2011*). Longitudinal-circumferential orientation is defined by organ shape (*Figure 4F*) and determines whether the organ grows longer or wider (*Mochizuki et al., 2014*; *Schnatwinkel and Niswander, 2013*; *Tang et al., 2011*). In epithelial development, well-understood mechanisms orient cell divisions for proper morphogenesis. In epithelial homeostasis, however, the existence of analogous orientation mechanisms is a subject of debate.

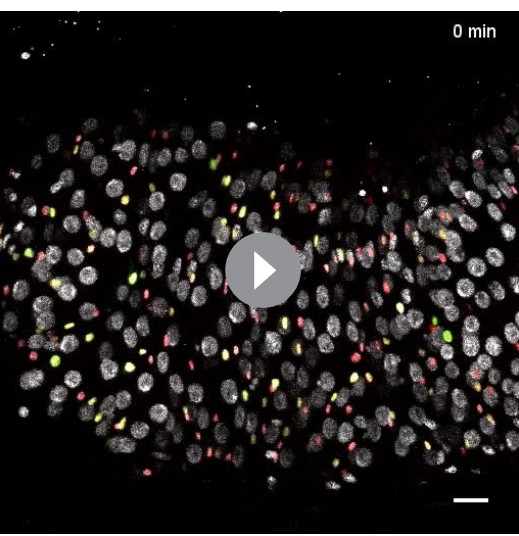

**Video 7.** Ten-hour movie of a 'fate sensor' midgut (*esgGal4, UAS-his2b::CFP, GBE-Su(H)-GFP:nls; ubi-his2av::mRFP*). See *Figure 2A–B*). Nuclei are distinguishable for four midgut cell types: stem cells (red pseudocolor), enteroblasts (yellow-green pseudocolor), enterocytes (gray, polyploid), and enteroendocrine cells (gray, diploid). Each time point is the projection of a confocal z-stack. Scale bar, 20 μm.
DOI: https://doi.org/10.7554/eLife.36248.014

To shed light on this topic, we investigated whether native mitoses in the midgut exhibited bias in horizontal-vertical or longitudinal-circumferential orientations. While performing these two analyses, we noticed an unexpected, third frame of reference: neighbor enteroblasts (*Figure 4I*). Below, we describe real-time mitotic orientation in these three reference frames.

First, we considered horizontal-vertical orientation (*Figure 4A–E*, *Figure 4—source data 1*). Horizontal-vertical orientations at cytokinesis ranged broadly (1.6°−72°), but were biased toward horizontal. Of 10 dividing cells, four were <5° and seven were <45° (*Figure 4B*). These findings are consistent with prior analyses of dividing cells in fixed midguts (*Goulas et al., 2012*; *Ohlstein and Spradling, 2007*).

Do horizontal-vertical orientations stay constant throughout division? Analyzing the same 10 cells from metaphase to telophase, we found that orientations were, as a group, also biased toward horizontal; of 41 measurements, seven were <5° and 33 were <45° (*Figure 4C*). Interestingly however, this stable, population-level trend belied the dynamic re-orientations of individual cells. Tracking single cells over time, we found that 8 of the 10 cells re-oriented by ≥15° at least once, and four re-oriented by ≥30° (*Figure 4D–E*; *Videos 12* and *13*). Re-orientations even occurred repeatedly during a single mitosis; 3 cells re-oriented by ≥15° two or three times. These frequent, sometimes dramatic, re-orientations were not triggered by peristaltic contractions as no temporal correlation was observed between the two types of events. The ability of mitotic stem cells to re-orient dynamically, a feature uniquely visible in live imaging, carries implications for how measurements of spindle angles in fixed midguts are interpreted.

Second, we considered longitudinal-circumferential orientation (*Figure 4F–H*). Measuring 38 cells at cytokinesis, we found that 20 cells were ≤45° and 18 cells were >45° (*Figure 4F–H*). Hence, longitudinal-circumferential orientations are unbiased.

The nature of our movies precluded us from examining a potential exception to this lack of bias: divisions at midgut compartment boundaries. Spradling and colleagues have reported that compartmentalization of the midgut into distinct, stereotyped regions is reinforced by clonal partitioning (*Marianes and Spradling, 2013*). Daughter cells generally remain in the same compartment as their mother stem cell, and stem-cell clones do not cross most compartment boundaries (*Marianes and Spradling, 2013*). As compartment boundaries are circumferential, a possible explanation for clonal partitioning is that boundary-localized divisions are oriented circumferentially. However, our movies lacked live boundary markers, so the small minority of divisions that may have occurred at boundaries could not be distinguished from the large majority of divisions that occurred within compartments. Further study will be needed to determine whether boundary-localized divisions represent a special case of circumferential bias.

Finally, we observed that a third, local reference frame formed when two enteroblasts flanked a dividing cell (*Figure 4I–K*). In this three-cell arrangement, divisions occurred nearly parallel to the two neighbor enteroblasts (4 of 18 divisions; *Figure 4J,K*, *Figure 4—source data 1*). By contrast, divisions had a broad range of orientations if only one neighbor enteroblast was present (11 of 18 divisions; *Figure 4K*). When trapped between two enteroblasts, daughter cells at cytokinesis hurled into and forcibly collided with the enteroblast nuclei (*Figure 4J*; *Video 14*, t=15–22.5 min). These observations suggest that physical contact between stem cells and enteroblasts is a spatial cue that orients the mitotic spindle.

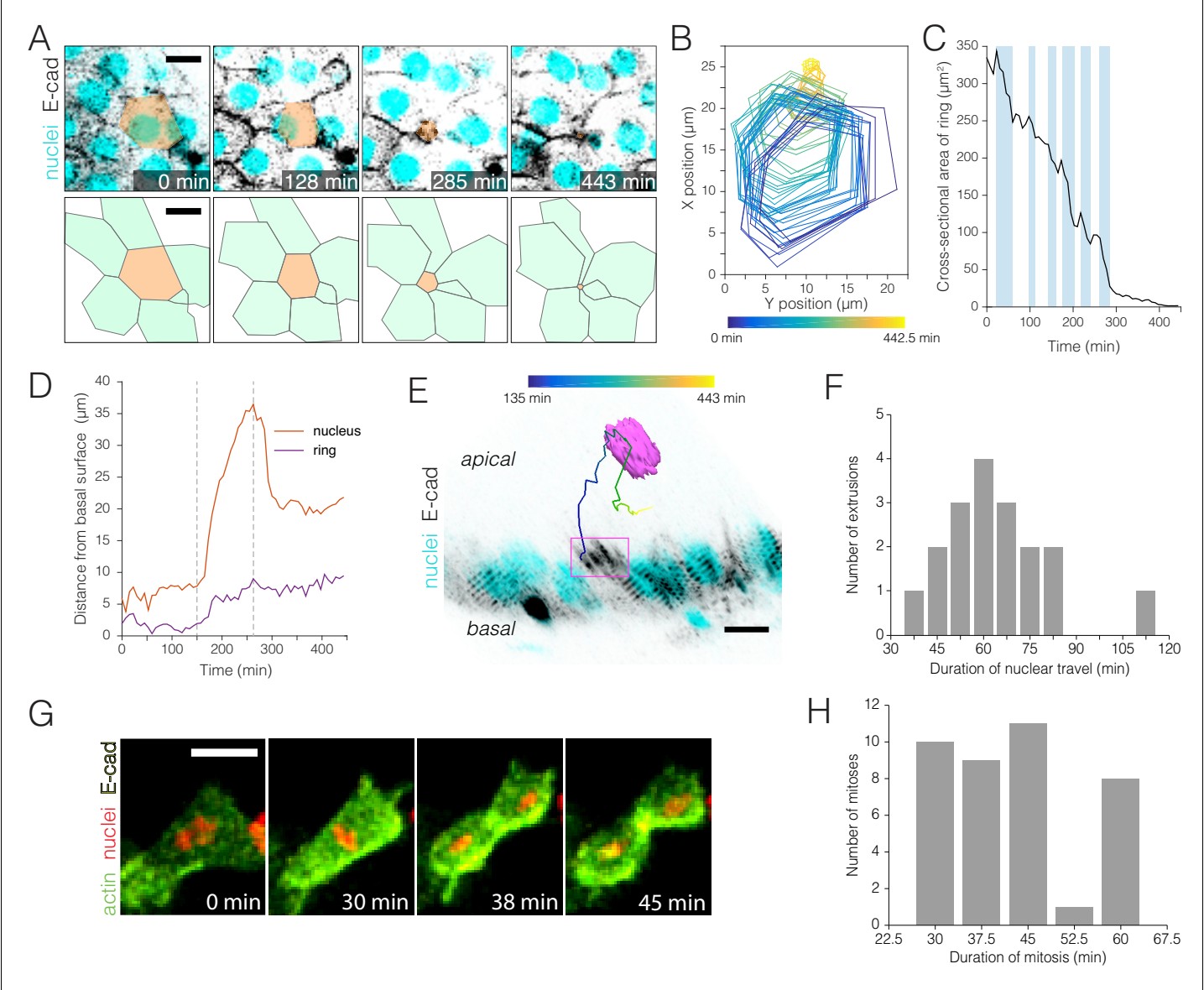

**Figure 3.** Real-time kinetics of enterocyte extrusion and stem cell mitosis. (**A–E**) Morphometric analysis of a single-enterocyte extrusion. (**A**) Time-lapse sequence (top) and schematic (bottom) showing a planar view of an extrusion event. The basal region of the extruding cell (tan pseudocolor) is outlined by a six-sided 'ring' of E-cadherin::YFP (inverted gray, *ubi-DE-cadherin::YFP*). Over time, the basal ring closes to a point, and the six neighbor cells (green in schematic) draw into a rosette. The time-lapse images are stack projections. Cyan (*ubi-his2av::mRFP*) labels all nuclei. See ***Video 8***. (**B**) Spatial 'footprint' of the E-cadherin::YFP ring in the epithelial plane over time (violet-yellow color scale). The ring remains six-sided throughout closure. (**C**) Ring closure occurs via ratcheted constrictions. During ring closure, pulses of constriction (shaded background) are interrupted by pulses of relaxation (unshaded background). See ***Figure 3—figure supplement 1***. (**D**) Kinetics of apical travel. Displacements of the junctional ring (purple) and the cell nucleus (red) are shown over time for the extrusion in (A). The ring (purple trace) advances incrementally via small, apical-and-basal movements. The nucleus (red trace) ejects rapidly into the lumen, then recoils. Apical nuclear travel starts at t = 150 min and ends at t = 263 min (dotted vertical lines). (**E**) Orthoview of the extrusion depicted in (A). The multicolored line shows the path of the nucleus over time (violet-yellow color scale). Magenta box denotes the E-cadherin::YFP ring, which is visible in this time point (t = 285 min) as a density of YFP at the apical surface. Inverted gray, E-cadherin::YFP (*ubi-DE-cadherin::YFP*); cyan, all nuclei (*ubi-his2av::mRFP*). See ***Video 9***. (**F**) Durations of apical nuclear travel for 18 single-enterocyte extrusions from six movies. Apical travel lasted 37–112 min with a mean ± standard deviation (SD) of 64 ± 18 min. (**G–H**) Kinetics of stem cell mitoses. (**G**) Time-lapse sequence of a mitotic event. Green, actin (*esg >LifeactGFP*); yellow, E-cadherin (*ubi-DE-cadherin::YFP*); red, nuclei (*ubi-his2av::mRFP*). Panels are partial stack projections of the basal epithelium. See ***Video 11***. (**H**) Durations of mitosis for 39 cell divisions from 11 movies. Mitoses lasted from 30 to 60 min with a mean ± SD of 43 ± 11 min. All scale bars, 10 µm.

DOI: https://doi.org/10.7554/eLife.36248.015

The following source data and figure supplements are available for figure 3:

*Figure 3 continued on next page*

*Figure 3 continued*

**Source data 1.** Raw data for *Figure 3F and 3H* and for mitotic index calculations.
DOI: https://doi.org/10.7554/eLife.36248.018
**Figure supplement 1.** Enterocyte extrusion occurs via ratcheted constriction of a basal junctional ring.
DOI: https://doi.org/10.7554/eLife.36248.016
**Figure supplement 1—source data 1.** Raw data for *Figure 3—figure supplement 1*.
DOI: https://doi.org/10.7554/eLife.36248.017

In summary, these analyses provide first views of how live stem cells orient their divisions within the midgut's tubular epithelium. They also reveal mitotic behaviors, such as frequent horizontal-vertical re-orientations, that are undetectable in fixed samples. Examining three reference frames, we found three orientation patterns. (1) Biased horizontal orientations. In future work, a crucial question will be whether, and if so how, horizontal orientations promote symmetric daughter fates (*Goulas et al., 2012*; *Guo and Ohlstein, 2015*; *Kohlmaier et al., 2015*; *Montagne and Gonzalez-Gaitan, 2014*; *Sallé et al., 2017*). (2) Unbiased longitudinal-circumferential orientations. This balanced distribution may help to maintain constant organ shape over time. (3) Local orientation by two enteroblasts. This unanticipated finding supports the notion that stem-cell–enteroblast adherens junctions, which are unusually pronounced (*Ohlstein and Spradling, 2006*), could orient the divisions of midgut stem cells, akin to other *Drosophila* stem cells (*Inaba et al., 2010*; *Le Borgne et al., 2002*; *Lu et al., 2001*). In our analysis, all mitoses occurred in the midgut's R4a-b (P1-2) region that is exposed by the cuticular window; whether divisions in other regions, or at region boundaries, behave similarly is an open question. Looking forward, these findings provide a basis for the direct investigation of midgut division orientation and for probing the relationship between orientation and fate.

## A quantitative threshold of Notch activation distinguishes stem cells and enteroblasts

Along with cell division and loss, cell differentiation is the third core behavior of tissue renewal. In the *Drosophila* adult midgut, differentiation in the enteroblast-enterocyte lineage is controlled by Delta-Notch. Delta ligand, which is expressed predominantly in stem cells, activates Notch receptors on stem (or stem-like) cells. At low levels, Notch activity is compatible with stemness, but at higher levels, it triggers enteroblast differentiation (*Bardin et al., 2010*; *Biteau and Jasper, 2014*; *Kohlmaier et al., 2015*; *Micchelli and Perrimon, 2006*; *Ohlstein and Spradling, 2006Ohlstein and Spradling, 2007*; *Perdigoto et al., 2011*; *Zeng and Hou, 2015*).

Fate sensor midguts (*Figure 2*) enable Notch activity to be measured live. *GBE–Su(H)-GFP:nls* provides a sensitive readout of Notch transcriptional activation (*de Navascués et al., 2012*; *Furriols and Bray, 2001*; *Guisoni et al., 2017*; *Housden et al., 2014*), while *ubi-his2av::mRFP* provides a stable reference signal. We used these two markers to establish a normalized metric of Notch activity. First, to account for differences between movies, we normalized the values of GFP and RFP intensities within a given movie to a 0-to-1 scale. Second, to account for tissue depth and other artifacts within a single movie, we used these normalized GFP and RFP intensities to calculate the ratio of GFP:RFP for each *esg >his2ab::CFP* progenitor cell at each time point. This two-part calculation of real-time GFP:RFP enables Notch activity to be compared over time and between cells, even in different movies.

We asked whether real-time GFP:RFPs are consistent with conventional indicators of enteroblast differentiation. The numeric values of GFP:RFPs, which ranged from 0.0 to 1.8, generally fit with subjective evaluations of GFP intensities (*Figure 5A–B*). In addition, population-level distributions of GFP:RFP were similar at different imaging depths, over time within a single movie, and across different movies. Furthermore, $esg^+$+ with large nuclei ($\geq$200 $\mu m^3$) often exhibited high GFP:RFPs, whwhereas cells with low GFP:RFP typically had small nuclei (*Figure 5B*). This association of high GFP:RFPs with large but not small enteroblast nuclei fits with prior observations that endoreplication is characteristic of late enteroblasts (*Jiang et al., 2009*; *Kohlmaier et al., 2015*; ; *Perdigoto et al., 2011*; *Xiang et al., 2017*; *Zhai et al., 2017*). Altogether, these findings support the use of GFP:RFPs as a metric of Notch activation.

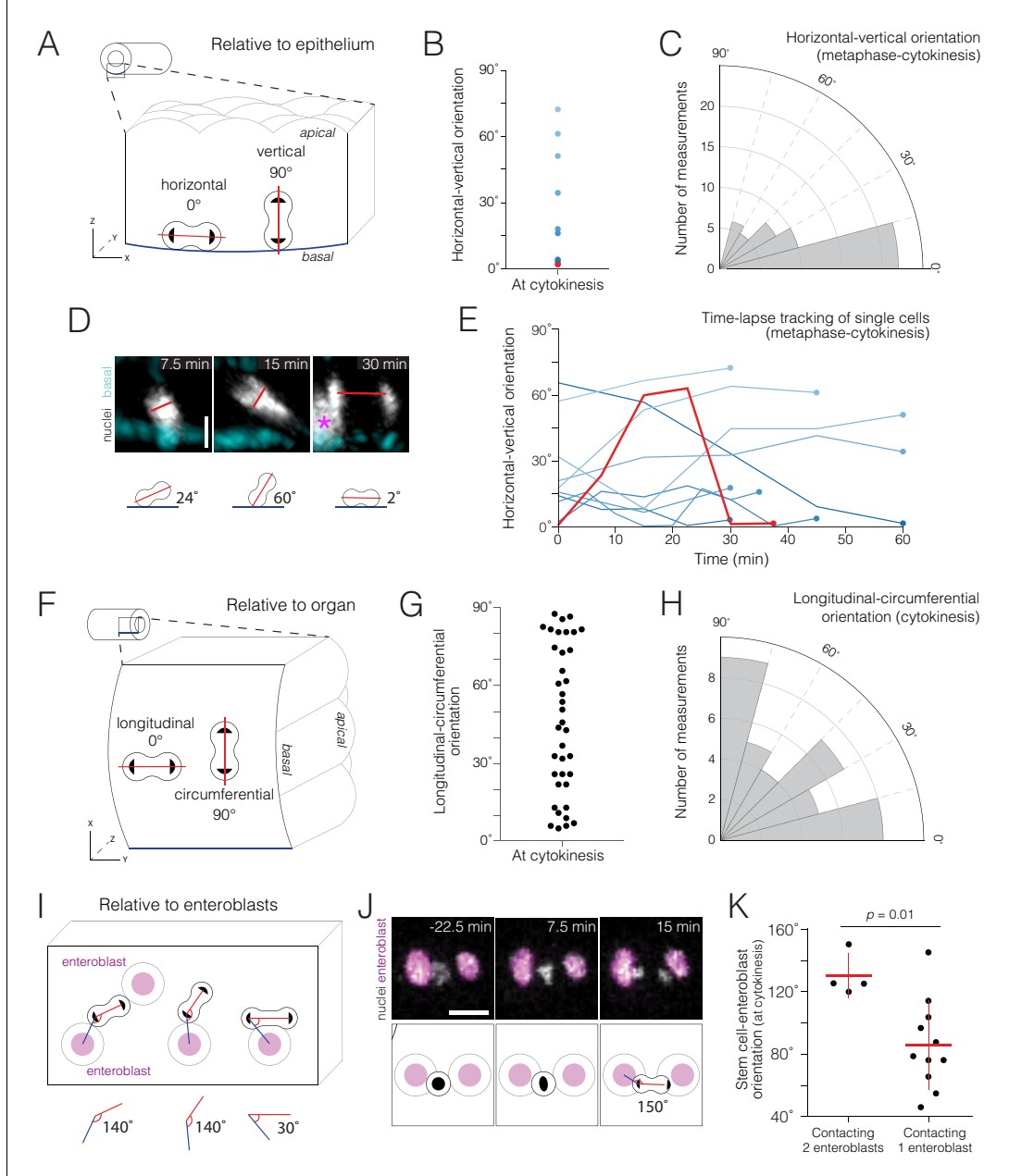

**Figure 4.** Real-time orientations of stem-cell divisions in three reference frames. (**A–E**) Horizontal-vertical orientations are horizontally biased. (**A**) Schematic of horizontal (0°) and vertical (90°) orientations. See *Figure 4—figure supplement 1*. (**B**) Live orientations of 10 dividing cells specifically at cytokinesis. The distribution is biased toward horizontal (<45°). The red point represents the cell in (D). (**C**) Live orientations of the same 10 cells throughout mitosis. Each measurement is the orientation of one mitotic cell at one time point, from metaphase to cytokinesis (n = 51 measurements). The distribution is biased toward horizontal (<45°). (**D**) Two re-orientations in a single mitosis. The red line shows the orientation of condensed chromatin (gray, *ubi-his2ab::mRFP*) relative to the basal basement membrane (cyan, Concanavalin A-Alexa-647). For clarity, in the 7.5 min and 15 min projections, a clipping plane was applied in the gray channel to exclude an enterocyte nucleus; this nucleus is marked by an asterisk at the left edge of the 30 min projection. Scale bar, 5 µm. See *Video 12*. (**E**) Mitotic cells frequently re-orient. Each line shows the horizontal-vertical orientations of a single mitotic cell over time. The 10 cells are the same as those in (B) and (C). All lines start at metaphase (t = 0 min) and continue until cytokinesis (t = 30–60 min). Time intervals were either 5, 7.5, or 15 min. Colors are the same as those in (B); the red line represents the orientation of the cell in (D). (**F–H**) Longitudinal-circumferential orientations are unbiased. (**F**) Schematic of longitudinal (0°) and circumferential (90°) orientations. (**G–H**) Live orientations of 38 dividing cells at cytokinesis. Longitudinal (≤45°) and circumferential (>45°) orientations are near-equal. (**I–K**) Divisions between two flanking enteroblasts align with the enteroblast-enteroblast axis. (**I**) Schematic of divisions contacting either two or one enteroblast(s). When two enteroblasts are present, the closer enteroblast is used for measurements (see 'Materials and methods'). (**J**) Division between two enteroblasts. Orientation is nearly parallel to the axis between the enteroblast nuclei (magenta, *GBE-Su(H)-GFP:nls*). Gray, stem cell and enteroblast nuclei

*Figure 4 continued on next page*

Figure 4 continued

(*esg >his2b::CFP*). Scale bar, 10 μm. See *Video 14*. (**K**) Live orientations of divisions with two or one flanking enteroblast(s). With two enteroblasts (n = 4 of 18 divisions), orientations are near-parallel to the enteroblast-enteroblast axis. With one enteroblast (n = 11 of 18 divisions), orientations are broadly distributed. Orientations were measured at cytokinesis. Means ± SD are shown. Mann-Whitney test, p=0.01.

DOI: https://doi.org/10.7554/eLife.36248.019

The following source data and figure supplement are available for figure 4:

**Source data 1.** Raw data for *Figure 4B, 4C, 4D, 4G, 4H, and 4K*.

DOI: https://doi.org/10.7554/eLife.36248.021

**Figure supplement 1.** Measurement of horizontal-vertical spindle orientation in space.

DOI: https://doi.org/10.7554/eLife.36248.020

Having validated this metric, we used GFP:RFPs to measure Notch activity in stem cells and enteroblasts. Genetic modulation of Notch signaling had revealed that stem and enteroblast identities are characterized by low and high activation (*Perdigoto et al., 2011*), but actual levels of Notch signaling were not quantified. We wondered whether GFP:RFPs could quantitatively distinguish stem cells and enteroblasts in real time.

Examining GFP:RFPs for all *esg*[+]+ cells in two fate sensor movies (29,102 values; 251 cells), we found that their distribution is—suggestively—bimodal. A local minimum at GFP:RFP=0.17 separates a sharp left peak (GFP:RFP=0.015) and a broad right peak (GFP:RFP=0.528) (*Figure 5C*, *Figure 5—source data 1*). An appealing interpretation of this bimodality is that the left peak represents stem cells and the right peak represents enteroblasts.

To test this interpretation directly, we cross-correlated Notch activity with mitotic behavior. Mitosis is near-exclusive to stem cells, so cells that went through mitosis during imaging were identifiable as stem cells, independent of GFP:RFP. The GFP:RFPs of these cells in the time points prior to their observed mitoses were used to create a 'benchmark' collection of GFP:RFPs from known stem cells (1,294 GFP:RFPs; 18 cells).

The benchmark collection of stem cell GFP:RFPs was compared to all progenitor GFP:RFPs. If the left peak of the *esg*[+]+ represents stem cells, then its GFP:RFP profile should resemble the profile of the 'benchmark' stem cells. Indeed, the two profiles nearly matched (*Figure 5C'*). Furthermore, 99.61% of GFP:RFPs for benchmark stem cells were less than the 0.17 threshold. This correspondence implies that stem cells populate the left peak and enteroblasts the right peak. Supporting this interpretation, the number of data points in the left and right peaks have a proportion of 4:3, which resembles the proportions of stem cells to enteroblasts that have been reported previously in fixed tissues (*Guisoni et al., 2017*; *O'Brien et al., 2011*).

On the basis of these findings, we conclude that GFP:RFP=0.17 marks a threshold level of Notch activation that functionally distinguishes stem cells from enteroblasts. The precise value of 0.17 is probably specific to our particular microscope and imaging parameters, and a different microscope system would require re-assessment of the enteroblast threshold through measurements of normalized GBE-Su(H)-GFP:nls intensities. Nonetheless, our findings argue that when Notch activity reaches a specific, critical level, cells transition from a stem-like state to an enteroblast state.

## Real-time kinetics of enteroblast transitions

A fundamental aspect of fate transitions is the time over which they occur. Fast transitions would allow cells to respond nimbly to acute challenges, whereas slow transitions would allow cells to receive and integrate a large number of fate-influencing signals. In this manner, the kinetics of fate transitions can define how an organ responds to changing external environments.

Midgut fate transitions have not been measured directly to date. For enteroblasts, an upper limit of two days can be inferred from observations that enteroblasts are present in stem-cell clones two days post-induction (*de Navascués et al., 2012*; *Ohlstein and Spradling, 2007*). In developing tissues, however, activation of Notch target genes can occur in minutes-to-hours (*Corson et al., 2017*; *Couturier et al., 2012*; *Gomez-Lamarca et al., 2018*; *Housden et al., 2013*; *Vilas-Boas et al., 2011*). This precedent raises the possibility that Notch-mediated enteroblast transitions in the midgut could occur over a period that is considerably shorter than two days.

To examine enteroblast transitions directly, we measured the rate of *GBE–Su(H)-GFP:nls* activation in movies of fate sensor midguts. A cell was scored as undergoing an enteroblast transition if

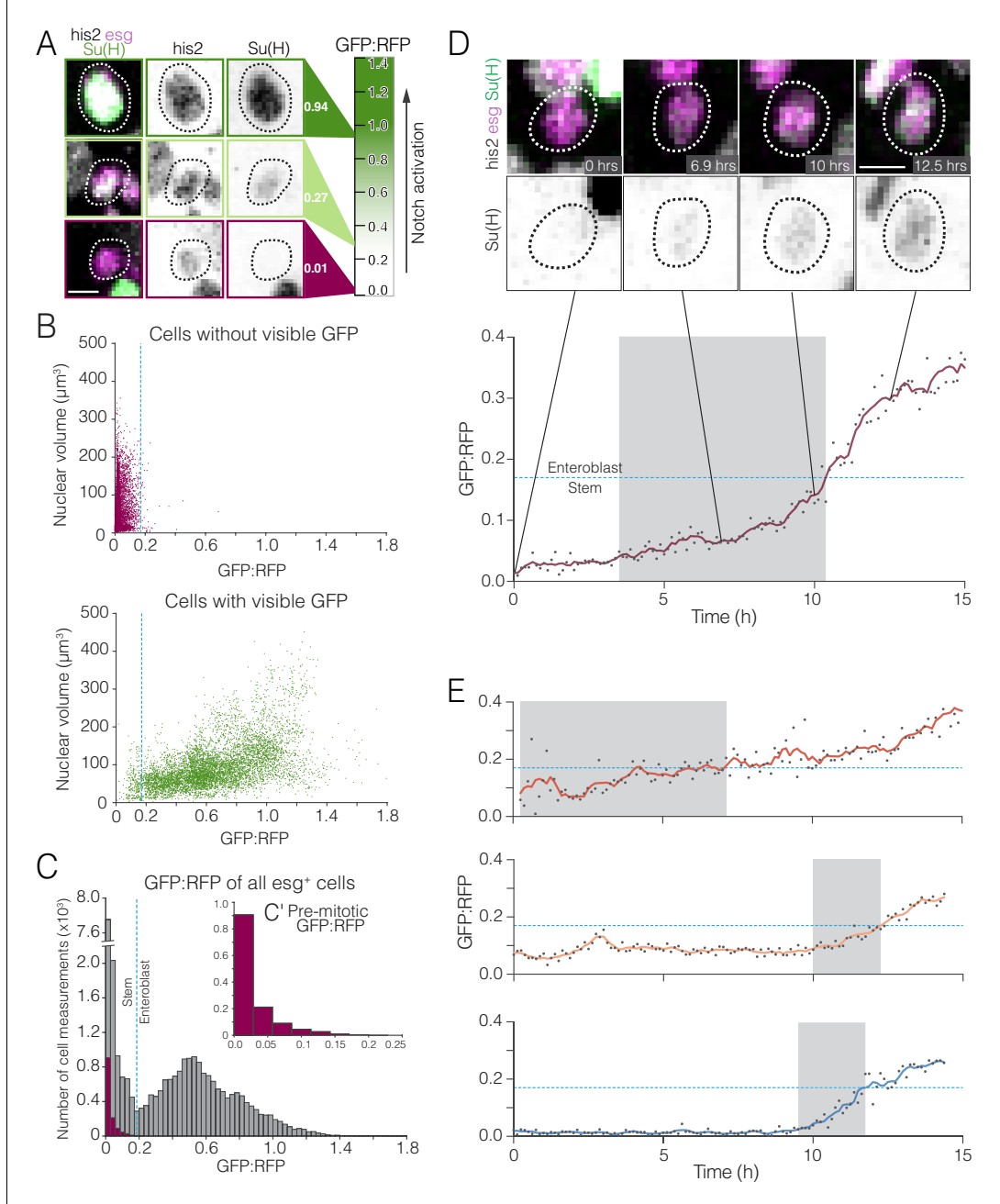

**Figure 5.** Whole-population and single-cell analyses of real-time Notch activation. (**A–C**) A threshold level of Notch activation distinguishes stem cells and enteroblasts. (**A**) Single-cell measurements of the Notch reporter *GBE-Su(H)-GFP:nls* from live movies. Cells additionally co-express *esg >his2ab:: CFP* (magenta) and *ubi-his2ab::mRFP* (gray). *GBE-Su(H)-GFP:nls* activation is quantified as GFP:RFP (see 'Materials and methods'). For the indicated cells, GFP:RFP = 0.94, 0.27, and 0.01, respectively. (**B**) GFP:RFP values correlate with visible GFP and nuclear volume. Progenitor (*esg+*) cells were scored by eye as either GFP-negative (top) or -positive (bottom). In cells without visible GFP, nearly all GFP:RFP values cluster between 0.0 and 0.2, and most nuclear volumes are small (<200 µm³). In cells with visible GFP, most GFP:RFP values are spread between 0.1 and 1.4, and large nuclear volumes (≥200 µm³), indicative of late enteroblasts, are associated with high GFP:RFPs. The blue dotted lines show the 0.17 enteroblast threshold from (C). (**C**) GFP: RFP values quantitatively distinguish stem cells and enteroblasts. Gray bars show real-time GFP:RFPs for all *esg+* cells in two movies (29,102 GFP:RFPs from 251 cells). Two peaks (GFP:RFP = 0.015, 0.528) are separated by a local minimum (blue dotted line; GFP:RFP = 0.17). Purple bars (**C'** inset) show real-time GFP:RFPs for 'benchmark' stem cells prior to an observed mitosis (1,294 GFP:RFPs from 18 pre-mitotic cells). The benchmark stem cell distribution matches the left peak of the *esg+* cells, and 99.6% of 'benchmark' GFP:RFPs are less than 0.17. Data in (B) and (C) are aggregated from two movies. (**D–E**) Stem-like cells transition to enteroblasts over multiple hours. (**D**) Real-time activation of *GBE-Su(H)-GFP:nls* reveals a transition from a stem-like to an enteroblast state. During a transition period lasting 6.9 hr (gray background), GFP:RFP increases from a baseline of ~0.049 at t = 3.5 hr

*Figure 5 continued on next page*

*Figure 5 continued*

to the enteroblast threshold of 0.17 (blue dotted line) at t = 10.4 hr. After the transition, GFP:RFP continues to increase and reaches 0.364 at t = 15.0 hr. *GBE-Su(H)-GFP:nls* shown in green (top) and inverted gray (bottom); *esg >his2ab::CFP*, magenta; *ubi-his2ab::mRFP*, gray. See *Video 15*. (E) Kinetics of three additional enteroblast transitions. Initial baseline GFP:RFPs are <0.17. GFP:RFPs increase from baseline to 0.17 during transition periods lasting from 2.3 to 6.9 hr (gray backgrounds: t = 0.3–7.1 hr (top), 10.0–12.4 hr (middle), 9.5–11.8 hr (bottom)). Initial and final GFP:RFPs are as follows: 0.058, 0.426 (top); 0.069, 0.281 (middle); 0.022, 0.257 (bottom). All cells in (D) and (E) were born before imaging started. Genotype in all panels: *esgGal4, UAS-his2b::CFP, Su(H)GBE-GFP:nls; ubi-his2av::mRFP*. All scale bars are 5 μm.

DOI: https://doi.org/10.7554/eLife.36248.022

The following source data is available for figure 5:

**Source data 1.** Raw measurements for *Figure 5*.
DOI: https://doi.org/10.7554/eLife.36248.023

GFP:RFP persistently increased from below to above the 0.17 threshold. From 95 cells with initial GFP:RFP <0.17 in two movies, five such transitions were identified. We analyzed kinetics for four of these, each of which occurred in a cell that was born before imaging started (*Figure 5D–E*, *Video 15*).

We found that enteroblast transitions (*Figure 5D–E*, gray background shading) occurred over multiple hours (2.4–6.9 hr)—faster than the 2 day upper limit implied by clones, and slower than the minutes observed in some other tissues. Higher initial GFP:RFPs did not correlate with shorter transition times. Whether the imaging protocol itself affected these kinetics is difficult to ascertain, but cumulative imaging stresses were probably not a factor as transitions that occurred later in imaging sessions did not have consistently slower or faster increases in GFP:RFP.

Intriguingly, we also observed *esg*[+]+ in which GFP:RFP fell from above to below 0.17 over several hours. These events might suggest that some nascent enteroblasts revert to a stem-like state, at least in terms of Notch activity. If so, then enteroblast specification, as marked by loss of mitotic capacity, occurs before commitment, in which terminal fate becomes irreversible. This two-step process is consistent with our observed, multi-hour timescale of Notch activation: during early transition stages, a prolonged period of low-level Notch activity probably results in prolonged expression of high-affinity Notch target genes. These targets, which are currently unknown, could serve to initialize a bistable switch that culminates in irreversible fate commitment (*Bray and Gomez-Lamarca, 2018*; *Ferrell and Xiong, 2001*). By lengthening the time between specification and commitment, slower activation may provide nascent enteroblasts with more opportunities to 'backtrack' if the tissue environment changes.

## Contacts between newborn siblings are variable and dynamic

In order to activate Notch, a prospective enteroblast must physically contact a Delta-expressing cell. In principle, newborn sibling cells would be ideally suited to engage in Notch-Delta interactions with each other (*Guisoni et al., 2017*): newborn cells express both Notch and Delta (*Bardin et al., 2010*; *Ohlstein and Spradling, 2007*), and cytokinesis leaves sibling cells juxtaposed. Sibling–sibling Notch activation requires that the two siblings stay in contact long enough to overcome the time delays inherent to Delta-Notch lateral inhibition (*Barad et al., 2010*; *Du et al., 2017*; *Guisoni et al., 2017*). However, fixed-gut studies of twin spot clones imply that after cytokinesis, some sibling pairs become separated (*O'Brien et al., 2011*). If sibling contacts can be transient, then the relationship between contact dynamics and Notch activation kinetics becomes crucial to enteroblast specification.

To investigate contact dynamics, we sought to visualize contacts between siblings directly by incorporating a membrane-localized YFP into our fate sensor line, which already contained nuclear-localized CFP, GFP, and RFP (*Figure 2A–B*). However, we were unable to parse the YFP signal without sacrificing sensitivity in the critical GBE-Su(H)-GFP:nls channel. As an alternative, we evaluated whether contact between siblings could be inferred from the distance separating their nuclei. To compare cell-cell contact and inter-nuclear distance directly, we used movies of midguts in which progenitor cell boundaries were visualized by LifeactGFP and nuclei by His2av::RFP (*Figure 6—figure supplement 1*, *Figure 6—figure supplement 1—source data 1*). This analysis revealed two strong correlations: progenitors with inter-nuclear distance <6.0 μm were nearly always in contact,

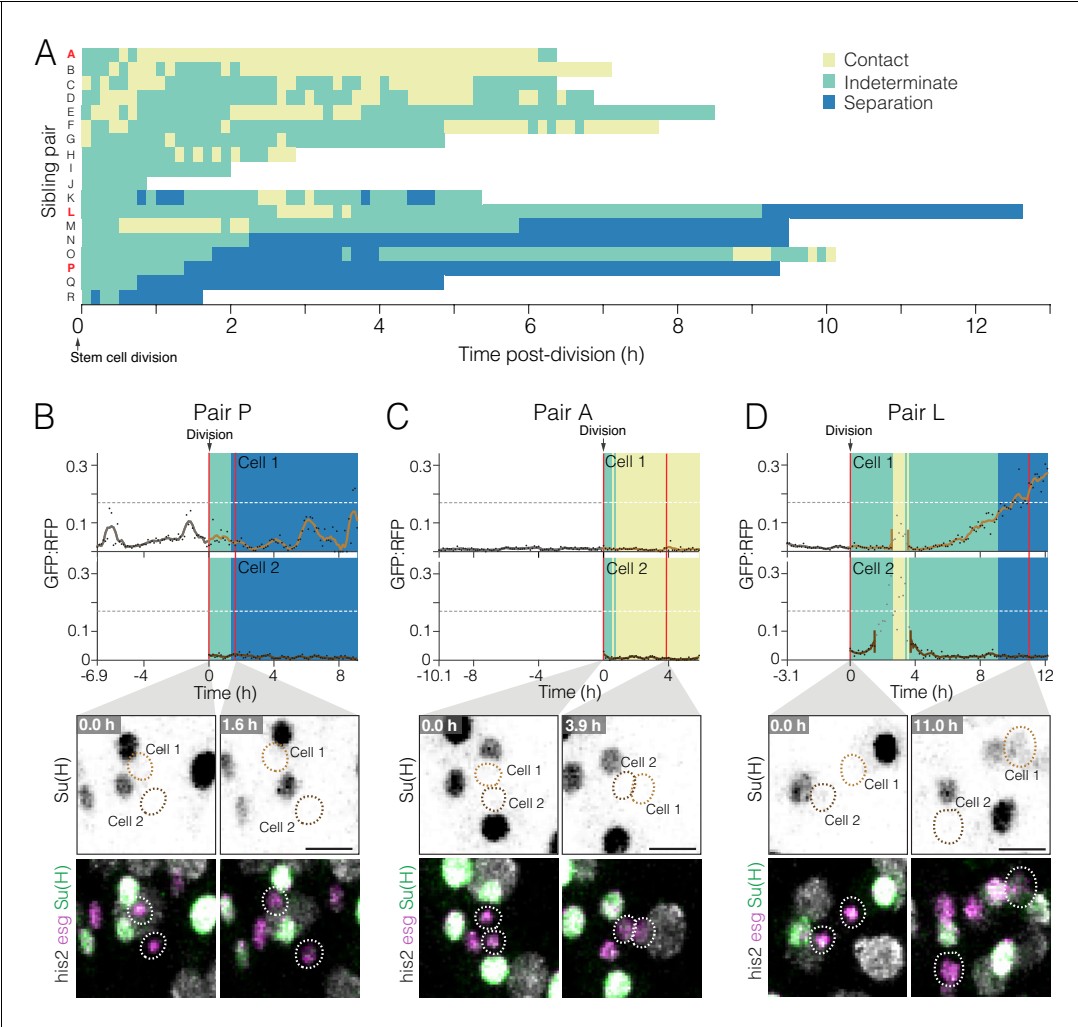

**Figure 6.** Dynamics of cell contact and Notch reporter activation in sibling cells after birth. (**A**) Contacts between newborn siblings are highly variable. Eighteen pairs of sibling cells (rows A–R) were tracked from birth (t = 0.0 hr) to the end of imaging. Color shows the likelihood of sibling–sibling contact based on inter-nuclear distance (**Figure 6—figure supplement 1**): yellow, inferred contact (inter-nuclear distance <6.0 µm); green, indeterminate (inter-nuclear distance = 6.0–15.5 µm); blue, inferred separation (inter-nuclear distance >15.5 µm). Pairs are ordered from highest A to lowest P contact. Pairs A, L, and P (red labels) are featured in (C), (D), and (B), respectively. (**B–D**) Contacts between siblings do not correlate with real-time *GBE-Su(H)-GFP:nls* activation. Graphs show real-time contact status (background colors same as A) and GFP:RFP ratios. Sibling birth is at t=0.0 h. Red vertical lines are the time points shown in the bottom images. (**B**) Low-contact pair P does not exhibit persistent activation of *GBE-Su(H)-GFP:nls*. (**C**) High-contact Pair A does not exhibit persistent activation of *GBE-Su(H)-GFP:nls*. (**D**) Indeterminate-low contact Pair L exhibits persistent activation of *GBE-Su(H)-GFP:nls* in one sibling. The Pair L siblings are probably in contact from t=2.6–3.6 h and are probably separated after t=9.1 h. Note that between t=2.6–3.5 h (Cell 1) and t=1.5–3.6 h (Cell 2), GFP:RFP measurements (grayed dots) are artifactually high because the two cells collide with a third cell, which was a mature enteroblast (Video 18). Because of the intimate proximity between the mature enteroblast and the Pair L siblings during the collision, the high GFP signal of the enteroblast bled over into the surfaces for Cells 1 and 2. The duration of artifactual bleed-over is indicated by gaps in the cells' interpolated GFP:RFP lines. Genotypes for all panels: *esgGal4, UAS-his2b:CFP, Su(H)GBE-GFP:nls; ubi-his2av::mRFP.* All scale bars are 10 µm. See Videos 16–18.

DOI: https://doi.org/10.7554/eLife.36248.024

The following source data and figure supplements are available for figure 6:

**Source data 1.** Raw data for **Figure 6**.
DOI: https://doi.org/10.7554/eLife.36248.027

**Figure supplement 1.** Comparison of cell-cell contact and inter-nuclear distance for live pairs of progenitor cells.
DOI: https://doi.org/10.7554/eLife.36248.025

**Figure supplement 1—source data 1.** Raw measurements for **Figure 6—figure supplement 1**.
DOI: https://doi.org/10.7554/eLife.36248.026

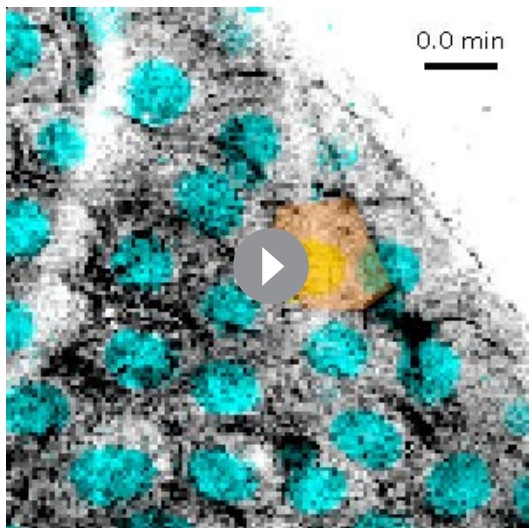

**Video 8.** Twelve-hour movie of a single-enterocyte extrusion. The epithelium is oriented with its basal surface toward the microscope objective and its apical surface further away. The basal region of the extruding enterocyte (orange pseudocolor at t=0, 127.5, 285, 442.5 min) is outlined by a 'ring' of E-cadherin::YFP. The ring closes down to a point from t=255–442.5 min. The intensity of the ring fluctuates during the first half of closure and becomes consistently bright during the second half. As the ring closes, neighboring cells draw into a rosette. Meanwhile, the nucleus of the extruding cell (yellow pseudocolor) starts to drop apically at t=150 min, hits its deepest luminal position at t=262.5 min, and recoils from t=262.5–307.5 min. Cyan, all nuclei (*ubi-his2av::mRFP*); inverted gray, E-cadherin (*ubi-DE-cadherin::YFP*). Each time point is the projection of a confocal stack. Scale bar, 10 µm.
DOI: https://doi.org/10.7554/eLife.36248.028

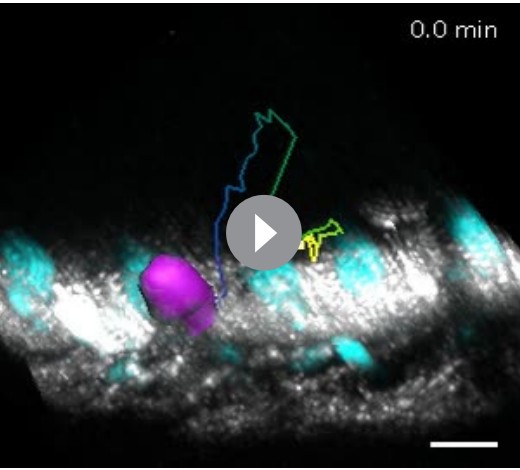

**Video 9.** Orthoview of extrusion shown in **Video 8**. The nucleus of the extruding enterocyte (magenta) ejects out of the epithelium (t=150–165 min) and penetrates into the lumen (t=165–265 min). It subsequently recoils and eventually comes to rest on the apical epithelium (t=263–443 min). The multicolored line shows the path of nuclear travel over time (violet-yellow color scale; see **Figure 3D** for legend). Cyan, all nuclei (*ubi-his2av:: mRFP*); gray, E-cadherin (*ubi-DE-cadherin::YFP*). Scale bar, 10 µm.
DOI: https://doi.org/10.7554/eLife.36248.029

and progenitors with inter-nuclear distance >15.5 µm were nearly always separated. Inter-nuclear distances between 6.0–15.5 µm did not correlate with either contact or separation. On the basis of these findings, we designated three classifications: inferred contact (inter-nuclear distance <6.0 µm), indeterminate (inter-nuclear distance $\geq$6.0 and$\leq$15.5 µm), and inferred separation (inter-nuclear distance >15.5 µm).

We used these classifications to examine the contact dynamics of sibling pairs in movies of fate sensor midguts. When analyzing 18 sibling pairs with known birth times, we found that they exhibited a broad diversity of contact behaviors (**Figure 6A**, **Figure 6—source data 1**). At one extreme were high-contact pairs, which generally stayed in place after cytokinesis (pairs A–C, **Figure 6A**). At the other extreme were low-contact pairs, which separated soon after cytokinesis (pairs N–R, **Figure 6A**). Eight of 18 pairs separated for at least one hour; two pairs separated and contacted repeatedly; and six pairs appeared to separate permanently. Notably, these dynamic, variable contact behaviors could not have been deduced from static images. Our observation that sibling cells routinely lose contact suggests that a substantial proportion of true sibling pairs may be missed by conventional fixed-gut assays that consider only contacting pairs.

Does sibling contact correlate with real-time Notch activation? To the contrary, both high- and low-contact siblings generally maintained low GFP:RFPs (**Figure 6B–C**, **Videos 16** and **17**). Of the 36 individual siblings that we tracked (**Figure 6A**), only one showed persistent activation of Notch (Cell 1 of Pair L, **Figure 6D**; **Video 18**). This particular cell was probably in contact with its sibling for at least one hour, perhaps longer, before its GFP:RFP began to increase (**Figure 6D**, t=4.0 hr). During the subsequent 6.2 hr, GFP:RFP climbed to the enteroblast threshold even after the two siblings probably lost contact (**Figure 6D**, t=9.1 hr). (Note that in **Figure 6D, a** collision of the **Figure 6D** siblings with a mature enteroblast caused GFP:RFP measurements to spike artifactually between t=2.6–3.5 hr (Cell 1) and t=1.5–3.6 hr (Cell 2). See **Figure 6D** caption and **Video 18**.) All other sibling cells

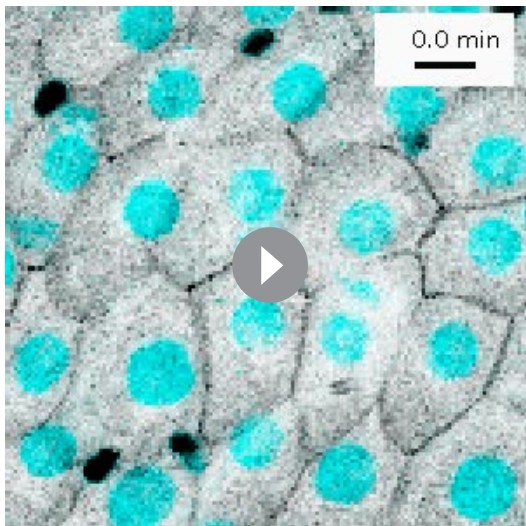

**Video 10.** Four-hour movie of an enteroendocrine cell extrusion. The epithelium is oriented with its basal surface toward the microscope objective and its apical surface further away. The basal region of the extruding cell (tan pseudocolor at t=0, 75 min) is outlined by a ring of E-cadherin::YFP (inverted gray signal at cell boundaries). The extruding cell is presumed to be enteroendocrine because it has a small, presumably diploid, nucleus and because it lacks expression of *esg*. (*esg >his2b::CFP* is inverted gray signal in nuclei.) The E-cadherin ring closes to a point over the period t=0–180 min. Meanwhile, the enteroendocrine cell nucleus drops apically from t = 0–143 min. Cyan, all nuclei (*ubi-his2av::mRFP*); inverted gray, E-cadherin (*ubi-DE-cadherin::YFP*) and stem/enteroblast nuclei (*esg >his2b::CFP*). Each time point is the projection of a confocal stack. Scale bar, 10 μm.
DOI: https://doi.org/10.7554/eLife.36248.030

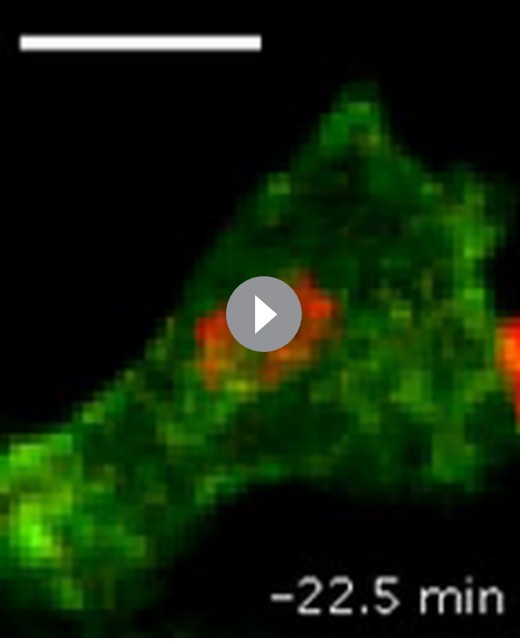

**Video 11.** Mitosis of a putative stem cell. Green, actin (*esg >LifeactGFP*); yellow, E-cadherin (*ubi-DE-cadherin::YFP*); red, nuclei (*ubi-his2av::mRFP*). Each time point is the partial projection of a confocal stack. Scale bar, 10 μm
DOI: https://doi.org/10.7554/eLife.36248.031

remained stem-like, with no persistent Notch activation, until the end of imaging. Had imaging continued, it is possible that additional siblings might have transitioned to enteroblasts. Unfortunately, the influence of sibling contacts on such hypothetical fates cannot be assessed. Nonetheless, a simple model in which sibling–sibling contact causes rapid, asymmetric Notch activation, akin to that of *Drosophila* sensory organ precursor cells (*Schweisguth, 2015*), is not supported by our live data.

## A delay in Notch activation?

Activation of Notch in new cells did not occur immediately, but rather several hours after birth. In *Figure 6D*, the Notch-activating sibling was 10 hr old when it reached the enteroblast threshold. In *Figure 5D–E*, cells had already been imaged for 7–12 hr when they reached the enteroblast threshold; as all of these four cells were born before imaging started, their elapsed times after birth were even longer. Because midgut cell divisions are asynchronous and stochastic, this post-birth delay in Notch activation would have been difficult to uncover without the time-resolved tracking of single cells.

Our finding that Notch activation is delayed raises at least three discussion points. First, it may explain why fewer enteroblast outcomes were observed using live versus fixed approaches. In our live movies, only 1 of 18 sibling pairs exhibited asymmetric Notch activation (*Figure 6D*). By contrast in prior, fixed studies that also used 2-day midguts, 20–30% of twin-spot sibling clones exhibited asymmetric, stem-enteroblast fates (*Chen et al., 2015*; *O'Brien et al., 2011*). A major difference in the two approaches is timescale: hours for live imaging, compared with days to weeks for twin-spot clones. With Notch activation delayed, some newborn cells that appeared stem-like during an hours-long movie might eventually have become an enteroblast or enterocyte in a days-old twin-spot

clone. In this manner, the delay in Notch activation may have caused enteroblast fates to be under-estimated in live movies.

Second, an important caveat is that we do not presently know whether Notch activation is comparably delayed in intact, unperturbed animals. Some potential confounding factors, such as the time required for GFP biosynthesis (*Balleza et al., 2018*; *Couturier et al., 2012*; *Housden et al., 2013*; *Kawahashi and Hayashi, 2010*; *Vilas-Boas et al., 2011*), involve timescales of minutes and are thus unlikely to be responsible for a delay of hours. However, because of technical challenges with pinpointing birth times for midgut cells in unperturbed animals, we cannot exclude the possibility that delayed activation is a consequence of the imaging protocol and not native physiological behavior.

Third, if delayed activation is physiological, then how exactly is it generated? And what are the consequences for dynamic fate control? One attractive notion is that a period of latency after birth could enable a cell to integrate a broad range of signals before choosing to either differentiate or self-renew. By allowing cells to process intrinsic and extrinsic fate signals fully, a latent 'waiting' period could ensure that individual cell fates are coordinated with overall organ needs.

## Conclusion

The *Drosophila* adult midgut is a premier model for organ renewal, but understanding the dynamics of renewal has been hampered by a lack of robust methodology for live imaging. Here, we have presented an imaging platform that captures the midgut in a near-native state within a live animal, yielding movies of exceptional visual quality and duration. In conjunction, we have described a pipeline for comprehensive, 4D movie analysis. We applied this pipeline to our movies for proof-of-principle analyses that corroborated fixed-tissue observations and uncovered new renewal behaviors. These novel findings ranged from descriptions of the time-resolved, single-cell dynamics of division orientation and apical extrusion to large-scale, population-level measurements of Notch activation. The ability to span cell- and tissue-level scales simultaneously over extended imaging periods opens the door to quantitative study of the spatiotemporal complexity of tissue renewal.

Despite these advances, our platform also has limitations. The positioning of the dorsal window restricts imaging to one organ region, so region-to-region comparisons cannot be made in real time. The media that covers the open window also dilutes the circulating hemolymph, which contains molecules that signal to midgut cells. Further refinement of our media formulation might help to restore division rates to native levels and to extend midgut viability beyond 16 hr. Similar improvements might be achieved with reduced exposure to laser light, for instance via a spinning disk set-up. Indeed, such enhancements will be needed to reach the paramount goal of capturing serial divisions of a single stem cell and tracing full lineages. Post-acquisition, our current registration algorithms cannot resolve movement-induced blurring in ~30% of raw movies; more sophisticated algorithms that correct for z-movements within a stack would render many of these movies analyzable.

In addition to midgut cell dynamics, our platform offers the opportunity to investigate other biological phenomena in the *Drosophila* abdomen. Animals ingest food during imaging, which enables real-time observation of events, such as colonization by ingested pathogens, that occur in the midgut lumen. By shifting the position of the cuticular window, the platform could also be adapted to study events in other abdominal organs, including ovary, testis, Malpighian tubule, and hindgut. We anticipate that long-term imaging of *Drosophila* adults will lead to a new, dynamic understanding of the cell and tissue behaviors that govern the form and physiology of mature organs.

## Materials and methods

**Key resources table**

| Reagent type (species) or resource | Designation | Source or reference | Identifiers | Additional information |
|---|---|---|---|---|
| Genetic reagent (*Drosophila melanogaster*) | esgGal4 | Kyoto DGGR | DGRC:112304; FLYB:FBti0033872; | FlyBase symbol: w[*]; P{w[+mW.hs] =GawB}NP0726/CyO |

*Continued on next page*

*Continued*

| Reagent type (species) or resource | Designation | Source or reference | Identifiers | Additional information |
| --- | --- | --- | --- | --- |
| Genetic reagent (*D. melanogaster*) | ubi-his2av::mRFP | Bloomington Drosophila Stock Center | BDSC:23650; FLYB:FBti0077846; RRID:BDSC_23650 | FlyBase symbol: w[*]; P{w[+mC]=His2Av-mRFP1}III.1 |
| Genetic reagent (*D. melanogaster*) | breathlessGal4, UAS-cyt-GFP | Other | | w; btl-Gal4, UAS-cytGFP shared by Mark Metzstein |
| Genetic reagent (*D. melanogaster*) | UAS-LifeactGFP | Bloomington Drosophila Stock Center | BDSC:35544; FLYB:FBti0143326; RRID:BDSC_35544 | FlyBase symbol: y[1] w[*]; P{y[+t*] w[+mC]=UAS-Lifeact-GFP}VIE-260B |
| Genetic reagent (*D. melanogaster*) | UAS-his2b::CFP | PMID: 24850412 | | w; UAS-his2b::CFP/(Cyo); + – shared by Yoshihiro Inoue |
| Genetic reagent (*D. melanogaster*) | GBE-Su(H)-GFP:nls | PMID: 22522699 | | w?; mw, GBE-Su(H)-GFPnls/(Cyo); Dr/TM6B – from (*de Navascués et al., 2012*) shared by Joaquin de Navascues |
| Genetic reagent (*D. melanogaster*) | act5c-spaghetti squash::GFP | PMID:12105185 | | w?; act5c-sqh::GFP; Dr/TM6C – shared by Denise Montell |
| Genetic reagent (*D. melanogaster*) | ubi-E-cadherin::YFP | PMID: 24855950 | | w; ubi-E-cadherin:: YFP; + – shared by Denise Montell |
| Chemical compound, drug | Concanavalin-A-Alexa647 | Invitrogen | Invitrogen:C21421 | 25 µg/ml final concentration |
| Chemical compound, drug | Sytox Green | ThermoFisher | Thermo Fisher:S7020 | 1 µM final concentration |
| Chemical compound, drug | SiR-tubulin | Cytoskeleton | Cytoskeleton: CY-SC002 | 0.5 µM final concentration |
| Chemical compound, drug | Human insulin | Sigma Aldrich | Sigma-Aldrich: I0516 | 100 µg/ml final concentration |
| Software, algorithm | Fiji | Other | RRID:SCR_002285 | StackRegfrom Arganda-*Arganda-Carreras et al., 2006b*; Correct 3D drift from *Parslow et al. (2014)*; Bioformats plugin |
| Software, algorithm | Bitplane Imaris | Other | RRID:SCR_007370 | Surpass module; Surface Recognition Wizard; Measurement Points tool |

## *Drosophila* husbandry

Fly stocks obtained from other sources:

esgGAL4 (Kyoto DGRC); *ubi-his2av::mRFP* (BL23650); *breathlessGal4, UAScyt-GFP* (Mark Metzstein); *UAS-LifeactGFP* (BL35544); *UAS-his2b::CFP* (Yoshihiro Inoue) (*Miyauchi et al., 2013*); *GBE-Su(H)-GFP:nls* (Joaquin de Navascués) (*de Navascués et al., 2012*); *act5c-spaghetti squash::GFP* (Denise Montell); and *ubi-E-cadherin::YFP* (Denise Montell) (*Cai et al., 2014*).

Generated stocks: act5c-spaghetti squash::GFP; ubi-his2av::mRFP esgGal4, UAS-his2b::CFP, GBE-Su(H)-GFP:nls; ubi-his2av::mRFP esgGAL4, UAS-his2b::CFP, GBE-Su(H)-GFP:nls/CyO; ubi-his2av::mRFP esgGal4, UAS-LifeactGFP; ubi-his2av::mRFP esgGAL4, GBE-Su(H)-GFP:nls; UAS-his2ab::mRFP.

Adult female *Drosophila* 2.5 days post-eclosion were used in all movies except *Video 3*, which used a female 7 days post-eclosion. Females were collected 0–4 hr post-eclosion, placed in vials with

males, and maintained at 25°C. Flies were fed a diet of standard cornmeal molasses food supplemented with yeast paste (1 g/1.4 ml $H_2O$).

## Fly mounts for extended live imaging on upright, inverted, and light-sheet microscopes

We designed three types of fly mounts that enable dorsal exposure of the midgut while stabilizing the live intact animal. For upright and inverted microscopes, our design is a modification of a previously published mount for the imaging of adult *Drosophila* brains (*Seelig et al., 2010*).

### Upright mount

To prepare the upright mount (*Figure 1—figure supplement 1A–B*), a stainless steel shim of 0.001 in thickness (Trinity Brand Industries, 612 H-1) was cut into 19 mm × 13 mm rectangles. From these rectangles, abdomen-sized cutouts were excised either by hand using an 18-gauge PrecisionGlide needle (Becton Dickinson, 305196) or by laser cutting using a micro laser cutting system (see *Figure 1—figure supplement 2* for CAD file). In addition, we prepared 60 mm Petri dishes (Fisher, FB0875713A) with a hole 10 mm in diameter drilled into the bottom. Each shim was glued onto the base of a 60 mm petri dish with clear silicone glue (DAP, 00688) and allowed to dry overnight.

The mount includes a feeder tube to provide the animal with liquid nutrients during imaging. We found that the feeder tube was essential for prolonged survival of the animal. Feeder tubes were made from 20 µl capillary tubes (Sigma-Aldrich, P0799), which were cut into 38 mm sections. Using a tungsten wire with a small hook bent at the end, a bit of cotton (Fisher Scientific, 22-456-880) was pulled through one end of the capillary tubing to form a feeder wick. Attachment of the feeder tube to the mount is described below.

A protective bottom chamber (*Figure 1—figure supplement 1A'*) enclosed the ventral side of the animal during imaging to prevent desiccation. To create the chamber, a 3 mm wide channel was drilled down the wall of a 35 mm Petri dish (Olympus Plastics, 32–103). Kimwipes (4-ply rounds, Fisher Scientific, 06–666) were cut and placed in the bottom of the humidity chamber to be soaked with water before use.

### Inverted mount

The inverted mount (*Figure 1—figure supplement 1C*) was similar to the upright mount, but used a glass bottomed Petri dish with two 1 mm spacers glued approximately 10 mm apart. Spacers were made from cut pieces of glass microscope slides (63720–05, Electron Microscopy Sciences) and were adhered with silicone glue. The same metal shim was used as with the upright mount, but was not affixed to the dish until the animal was glued and its gut stabilized (see below). Once the animal was prepared, the mounting shim was positioned with animal's dorsal side toward the glass bottom of the dish and glued to the spacers using KWIK-SIL adhesive silicone glue (World Precision Instruments, 60002).

### Light-sheet mount

Zeiss light sheet systems require a submersible chamber. To create such a chamber, we used the barrel of a 1 ml syringe in which one end was open to air (*Figure 1—figure supplement 1D*). A 5 mm x 8 mm square was cut into the side of the syringe barrel, and a metal shim with an abdominal cutout was affixed to the square window using silicone glue. A second window was cut into the opposite side of the barrel to provide physical access for mounting the animal and feeder tube inside. Once the animal and feeder tube were in place, the access window was sealed using a second metal shim and KWIK-SIL glue (World Precision Instruments, 60002). The bottom end of the syringe was sealed with dental wax (Surgident, 50092189) and the barrel was submerged in media in the Zeiss sample chamber. In this manner, the midgut was bathed in media during imaging while the animal's head and ventral surface remained in an open-air environment.

## Composition of imaging media and agarose

Media for midgut imaging was based on prior recipes for *Drosophila* organ culture ex vivo (*Morris and Spradling, 2011*; *Zartman et al., 2013*). Schneider's Insect Medium (Sigma-Aldrich, S0146) was supplemented with 5% FBS (Sigma-Aldrich, F4135), 5% fly extract (DGRC) (*Currie et al.,*

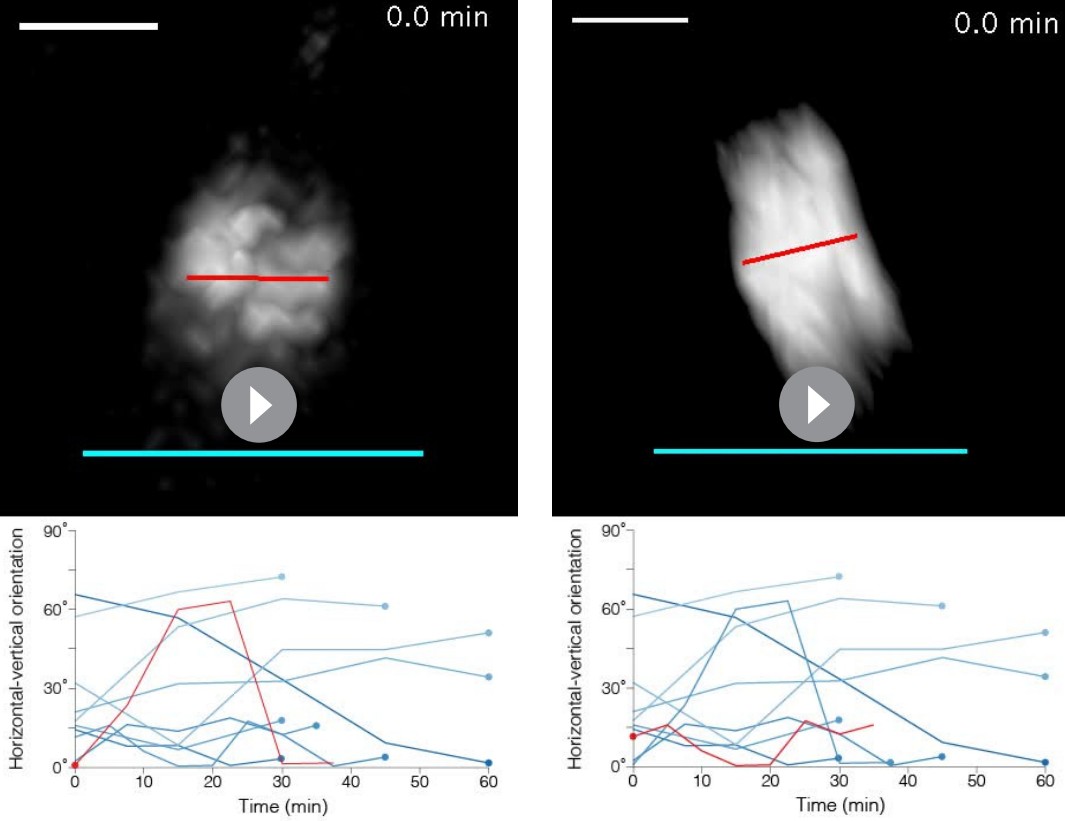

**Video 12.** Orthoview of a mitosis with two horizontal-vertical re-orientations. The first re-orientation occurs between metaphase (24° at 7.5 min) and anaphase (60° at 15 min). The second re-orientation occurs between anaphase (62° at 22.5 min) and telophase (2° at 30 min). Gray channel, condensed chromatin (*ubi-his2ab:: mRFP*). The red line indicates the spindle axis. The cyan line indicates the basal plane, as revealed by the basement membrane stain Concanavalin A-Alexa 647 (not shown). At each time point, the mitotic cell is shown as an orthogonal projection from the vantage of a plane that is parallel to the spindle axis and normal to the basal epithelial plane. For clarity, a clipping plane was applied in the gray channel to exclude an adjacent enterocyte nucleus. Scale bar, 5 μm.
DOI: https://doi.org/10.7554/eLife.36248.032

**Video 13.** Orthoview of a second mitosis with two horizontal-vertical re-orientations. The top panel shows condensed chromatin of the dividing cell (*ubi-his2ab:: mRFP*). The red line indicates the spindle axis. The cyan line indicates the basal plane, as revealed by the basement membrane stain Concanavalin A-Alexa 647 (not shown). The bottom panel reproduces the graph from *Figure 4E*, with the time-resolved orientations of this particular cell in red. The first re-orientation occurs during metaphase (from 16° at 5 min to 0° at 15 min). The second re-orientation occurs between metaphase (1° at 20 min) and anaphase (18° at 25 min). Scale bar, 5 μm.
DOI: https://doi.org/10.7554/eLife.36248.033

*1988*), 100 μg/mL human insulin (Sigma-Aldrich, I0516) and 0.5% penicillin-streptomycin (Invitrogen, 15140). (Without antibiotics, the imaging media became visibly contaminated after several hours of imaging.) Insulin was added fresh on the day of imaging.

Low-melting point agarose was used to stabilize the midgut during imaging. To make the agarose, 2-Hydroxyethylagarose (Sigma-Aldrich, A4018) was mixed with Schneider's Insect Medium to make a 6% w/v slurry. The slurry was heated to 65°C to melt the agarose, mixed thoroughly and separated into 25 μL aliquots that were stored at 4°C. Prior to imaging, aliquots were heated to 65°C, mixed 1:1 with a 2x concentration of imaging media, and applied to midguts as described below.

Where indicated in the text, various fluorescent dyes were added to the imaging media to visualize particular cell structures or conditions. (1) Concanavalin A-Alexa647 (Invitrogen, C21421) was used to stain the basement membrane. A stock solution of 5 mg/mL Concanavalin A in 0.1 M sodium bicarbonate was diluted 1:200 in 1x imaging media to obtain a final working concentration of 25 μg/

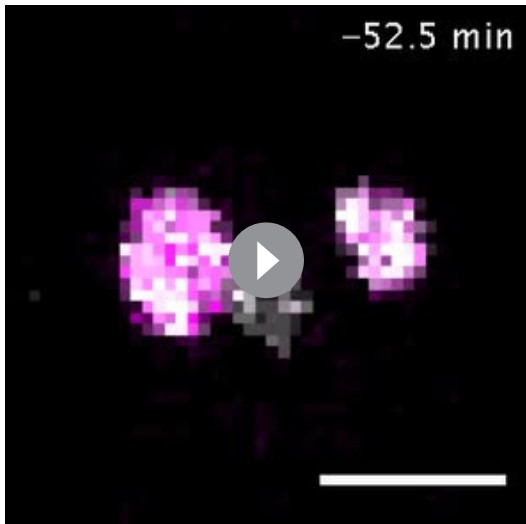

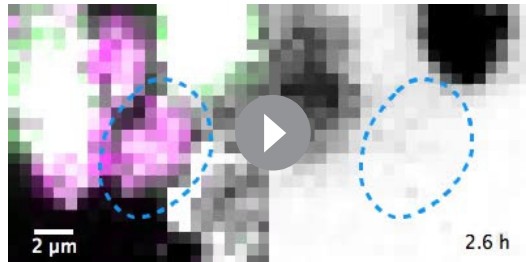

**Video 14.** Division of a stem cell that contacts two enteroblasts. Division orientation aligns with the axis between the two enteroblast nuclei (magenta, *GBE-Su (H)-GFP:nls*). At cytokinesis (t=15–22.5 min), the new daughter nuclei hurl into the enteroblast nuclei, which recoil in response. Gray, stem cell and enteroblast nuclei (*esg >his2b::CFP*). Each time point is the partial projection of a confocal stack. Scale bar, 10 μm.
DOI: https://doi.org/10.7554/eLife.36248.034

**Video 15.** Real-time enteroblast transition. In the incipient enteroblast (blue dotted circle), *GBE-Su(H)-GFP:nls* is initially undetectable (GFP:RFP=0.014 at t=0.0 hr). Over time, its GFP intensity increases, eventually reaching the enteroblast threshold (GFP: RFP=0.18 at t=10.5 hr). See *Figure 5D*. Left video: green, *GBE-Su(H)-GFP:nls.*; magenta, stem cell and enteroblast nuclei (*esg >his2b::CFP*); gray, all nuclei (*ubi-his2av::mRFP*). Right video: inverted gray, *GBE-Su (H)-GFP:nls*. Each time point is the partial projection of a confocal stack. Scale bar, 2 μm.
DOI: https://doi.org/10.7554/eLife.36248.035

mL. A drop of this media was placed on the dorsal cuticle prior to cutting the cuticle window. Agarose and subsequently added media did not contain Concanavalin A. (2) SiR-Tubulin (Cytoskeleton, CY-SC002) was used to stain microtubules. SiR-Tubulin was added to 1x imaging media for a final working concentration of 0.5 μM. A drop of this media was placed on the dorsal cuticle prior to cutting the cuticle window. Agarose and subsequently added media did not contain SiR-tubulin. (3) Sytox Green (ThermoFischer, S7020) was used to mark dying cells. A stock solution of 5 mM Sytox Green in dimethyl sulfoxide (DMSO) was diluted 1:5000 in 1x imaging media to a final concentration of 1 μM. Sytox-Green-containing media was placed over the agarose immediately prior to imaging.

## Animal preparation

A narrated tutorial video (*Video 1*) provides step-by-step instructions for mounting the animal and exposing the midgut. Wings were broken off near the hinge using forceps. Flies were placed in a microfuge tube on ice for at least 1 hr before being glued dorsal side down to the fly mount (*Figure 1—figure supplement 1*) using KWIK-SIL glue. To optimize access to region R4 of the gut, the fly was tilted toward its left side when glued to the mount. For long-term survival of the animal, its right-side spiracles were kept open (*Video 1*). After the glue had dried, the feeder tube, filled with 10% sucrose (w/v) in H$_2$O, was secured to the fly mount with dental wax (Surgident, 50092189) and positioned such that the cotton wick was in reach of the animal's proboscis. The protective bottom chamber was attached to the bottom of the Petri dish with masking tape (*Figure 1—figure supplement 1A'*).

To expose the midgut, a window was cut into the dorsal cuticle as follows. First, a drop of imaging media was placed onto the dorsal cuticle. Next, portions of cuticular segments A1 and A2 were excised using Dumont #55 forceps. In the majority of animals, this excision exposed the looped midgut region R4a-b/P1-2 (*Buchon et al., 2013b*; *Marianes and Spradling, 2013*). The loop was gently coaxed using forceps to protrude slightly out of the window. The imaging media was temporarily removed, and a drop of the agarose mixture (described above) was applied to the exposed loop and allowed to solidify. Once the agarose had hardened, a drop of media was added on top of the agarose to avoid desiccation. Between steps, the setup was placed on ice to minimize animal movement. The bottom of a 60 mm Petri dish was used to cover the mounted animal until ready for placement on the microscope.

We explored the alternative of cutting a window in the ventral, rather than dorsal, cuticle, but we found a ventral window to be unsuitable for long-term viability. When animals were mounted ventrally, the feeder tube could not be positioned correctly and the spiracles could not be left unoccluded. In addition, the pliable nature of the ventral cuticle often resulted in unpredictable tearing during cutting.

## Microscopy

An upright Leica SP5 multi-photon confocal microscope and a 20x water immersion objective (Leica HCX APO L 20x NA 1.0) were used to acquire the movies that were analyzed in this study. The microscope was controlled via a Leica CTR6500 controller card on a Z420 (Hewlett Packard) workstation with 16 GB memory and a Xeon CPU E5-1620 (Intel) running Windows 7 Pro and the Leica Application Suite: Advanced Fluorescence (LAS AF, v.2.7.3.9723). In addition, an inverted Leica SP8 confocal microscope with a 20x oil immersion objective (Leica HC PL APO IMM CORR CS2 NA 0.75) and a Zeiss light sheet Z.1 with a 20x water immersion objective (Zeiss light sheet detection optics 20X NA 1.0) were used to test the fly mounts for these microscope setups. For upright and inverted setups, a humidity box was assembled around the lens and the specimen to prevent desiccation (*Figure 1—figure supplement 1B*). The humidity box was formed from a pipette box lid with a hole for the lens and an inlet tube for humidified air. The box was connected to a 500 ml Pecon humidification bottle containing distilled water, and humidified ambient air was piped into the box via a Pecon CTI-Controller 3700. In addition, for upright setups, the Kimwipes in the protective bottom chamber were saturated with distilled water. For upright setups, 2–3 ml of imaging media were added to the sample, spanning the distance between the exposed midgut and the water immersion objective. Movies were captured at room temperature (20–23°C). Confocal stacks were acquired with a z-step of 2.98 µm and typically contained ~35 slices. For ex vivo imaging (*Figure 1—figure supplement 3A*), the upright Leica SP5 multi-photon confocal microscope was used with a 20x oil immersion objective (HC PL APO 20x/IMM N.A. 0.7).

## Ex vivo imaging

To provide a positive control for Sytox Green dead-cell staining (*Figure 1—figure supplement 3A*), we generated dying midgut cells by dissecting midguts and culturing them ex vivo for 2.5 hr in

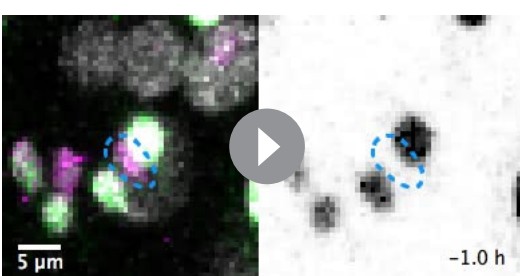

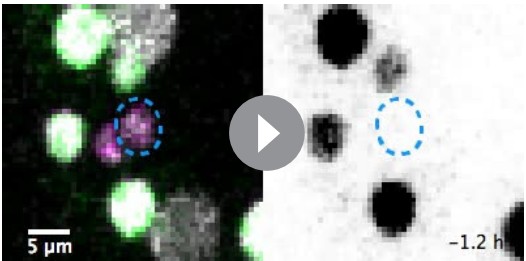

**Video 16.** A low-contact sibling pair (Pair P; *Figure 6A, B*) does not activate *GBE-Su(H)-GFP:nls*. Following their birth at t=0.0 hr, the two siblings move apart and have probably lost contact by t=1.4 hr (inter-nuclear distance >15.5 µm; c.f. *Figure 6—figure supplement 1*). The mother stem cell is indicated by the blue dotted circle at t=−1.0 hr; the two siblings are indicated by the two blue dotted circles at t=0.0 and t=9.2 hr. No GFP expression is apparent in either sibling. Left video: green, *GBE-Su(H)-GFP:nls*; magenta, stem cell and enteroblast nuclei (*esg >his2b::CFP*); gray, all nuclei (*ubi-his2av::mRFP*). Right video: inverted gray, *GBE-Su(H)-GFP:nls*. Each time point is the partial projection of a confocal stack. Scale bar, 5 µm.
DOI: https://doi.org/10.7554/eLife.36248.036

**Video 17.** A high-contact sibling pair (Pair A, *Figure 6A,C*) does not activate *GBE-Su(H)-GFP:nls*. Following their birth at t=0.0 hr, the two siblings probably remain in contact (inter-nuclear distance <6.0 µm; c.f. *Figure 6—figure supplement 1*) for at least 6.0 hr. The mother stem cell is indicated by the blue dotted circle at t=−1.2 hr; the two siblings are indicated by the two blue dotted circles at t=0.0 and t=6.0 hr. No GFP expression is apparent in either sibling. Left video: green, *GBE-Su(H)-GFP:nls*; magenta, stem cell and enteroblast nuclei (*esg >his2b::CFP*); gray, all nuclei (*ubi-his2av::mRFP*). Right video: inverted gray, *GBE-Su(H)-GFP:nls*. Each time point is the partial projection of a confocal stack. Scale bar, 5 µm.
DOI: https://doi.org/10.7554/eLife.36248.037

Schneider's Medium that contained 1 µM Sytox Green (ThermoFisher, S7020). An 8-well Secure-Seal spacer sticker (ThermoFisher S24737) was used to form 'wells' on a microscope slide (Fischer Electronic Microscopy Sciences 63720–05). One midgut and 7 µl of Sytox-containing medium were placed in each well. A coverslip (Fisher Scientific 12-545-81) was placed on top of each well. Midguts were imaged immediately and 2.5 hr after mounting.

## Movie registration and cell masking in Fiji

After acquisition, movies were processed on a Mac Pro computer (OS X 10.8.5) with a 3.2 GHz quad-core Intel Xeon processor and 20 GB memory. LIF files (*.lif) from LAS AF were uploaded into Fiji as a hyperstack for registration. To correct for X-Y drift, movies were converted to RGB files and processed with the Fiji plugin StackReg (*Arganda-Carreras et al., 2006a*). To correct for global volume movements, movies were processed with the Fiji plugin Correct 3D Drift (*Parslow et al., 2014*).

After registration, movies were evaluated for midgut viability. Viability was deemed to be compromised when any one of the following behaviors were observed: (1) wholesale dimming of fluorescent signals; (2) enterocyte extrusions *en masse* (>12 cells at once); (3) loss of ordered enterocyte packing; (4) appearance of multiple pyknotic nuclei; or (5) widespread flattening and spreading of progenitor cells. Once any of these

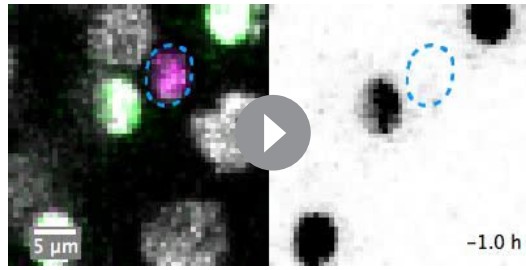

**Video 18.** A sibling pair exhibits asymmetric Notch activation (Pair L, *Figure 6A,D*). Following their birth at t=0.0 hr, the two siblings are probably in contact from t=2.6–3.6 hr, in indeterminate contact from t=3.6–9.0 hr, and separated after t=9.0 hr. The mother stem cell is indicated by the blue dotted circle at t=−1.0 hr. The two siblings are indicated by the two blue dotted circles at t=0.0 and 12.2 hr. A single blue dotted circle at t=10.2 hr indicates when the Notch-activated sibling crosses the enteroblast threshold (GFP:RFP=0.17; c.f. *Figure 6D*). This sibling exhibits nascent GFP signal at 4.0 hr and increases in GFP intensity for the rest of the movie. The other sibling does not exhibit detectable GFP, but from t=1.5–3.6 hr, it collides with a high-GFP enteroblast (orange dotted circle), which causes GFP signal to 'bleed over' in the GFP:RFP analysis (*Figure 6D*). Left video: green, *GBE-Su(H)-GFP:nls*; magenta, stem cell and enteroblast nuclei (*esg >his2b:: CFP*); gray, all nuclei (*ubi-his2av::mRFP*). Right video: inverted gray, *GBE-Su(H)-GFP:nls.* Each time point is the partial projection of a confocal stack. Scale bar, 5 µm.

DOI: https://doi.org/10.7554/eLife.36248.038

criteria were observed, the remainder of the movie was not used for further analysis.

For identification of individual cells, intensity thresholding of the ubiquitously expressed nuclear marker *ubi-his2av::RFP* was used to segment out individual cell nuclei. In fate sensor movies (see *Figure 2*), intensity thresholding for CFP (stem cells and enteroblasts; *esg >his2b::CFP*) and GFP (enteroblasts; *GBE-Su(H)-GFP:nls*) was applied to define masks of nuclei within each channel. Using Fiji's image calculator function, these masks were used to isolate the His2av::RFP-marked nuclei for individual cell types. Specifically, after registration (*Figure 1E*), channel masks were generated in Image J to isolate stem cells digitally (CFP channel minus GFP channel), enteroblasts (GFP channel), and mature enterocytes and enteroendocrine cells (RFP channel minus CFP channel). To isolate enterocytes and enteroendocrine cells, whose populations were defined by the absence of *esg >his2av::CFP*, the His2b::CFP-marked progenitor nuclei were eliminated using the subtraction function in Fiji's image calculator.

Masked nuclei for these three populations were added to the raw hyperstack as unique channels for use in Bitplane Imaris (see below). The two types of mature cells, enterocytes and enteroendocrine cells, were then distinguished by a size filter in Imaris; nuclei $\leq$ 113 µm$^3$ were classified as enteroendocrine, whereas nuclei > 113 µm$^3$ were classified as enterocyte. To maintain metadata structure for 4D Imaris analysis (see below), final movies were exported to OME-TIFF format (*.ome.tif) using the BioFormats plugin in Fiji.

## Cell identification and tracking in Imaris

To perform cell tracking, Fiji processed stacks of midgut movies were opened in Bitplane Imaris from OME-TIFF format (*.ome.tif) files. Once converted to an Imaris *.ims file, the 4D volumes were

visually inspected using the Surpass module to verify the accuracy of image processing and file conversion. Cell segmentation was then performed by applying the Surface Recognition Wizard module to the masked cell channels generated in Fiji (see above). Final products were visually compared to raw channels to confirm cell-type recognition.

Cell surfaces were tracked using the Brownian motion tracking algorithm. Automated Imaris tracking accurately identified ~70–90% of cells, depending on the frequency of organ movements in the movie. Visual inspection was used to correct errors. Once cell recognition was verified for all cells and time points, individual cell statistics were exported as either a Microsoft Excel file or a comma-separated-value file. The data were then imported into Mathematica or MATLAB for quantitative analysis.

For *Figures 5* and *6* and their associated data, a modification of the above protocol was used. To identify cells that transitioned over time from a stem-like state (GFP:RFP $\leq$0.17) to an enteroblast state (GFP:RFP >0.17), cells expressing *esg >his2b::CFP* were identified in Imaris. Their GBE-Su(H)-GFP:nls intensities and nuclear volume were determined at each time point. Cells exhibiting increasing GFP intensities were identified and selected for further analysis.

## Spatiotemporal analyses of enterocyte extrusion

### Analyses of the E-cadherin::YFP ring

Extruding enterocytes were identified by visual inspection. To measure the dynamics of the E-cadherin::YFP ring (*Figure 3B–E*), Fiji-processed movies were opened from OME-TIFF files (*.ome.tif) in Bitplane Imaris and viewed as 3D volumes using the Surpass module. Vertices of the E-cadherin::YFP ring that outlined the extruding cell were identified by visual inspection at each movie time point. The Measurement Points tool was used to place a polygon-mode measurement point at each vertex. In addition, a plane representing the basal epithelium was defined by selecting three Measurement Points on the basal epithelial surface underlying the extruding cell. To identify the position of the basal surface, we used either the basement membrane stain Conconavalin-A-Alexa647 or the background fluorescence of the cytoplasm of enterocytes when movies were digitally overexposed. The spatial coordinates of all these measurement points were exported as comma-separated values and imported into MATLAB.

To map the 'footprint' of the ring in the epithelial plane (*Figure 3B*), the coordinates of the ring vertices were connected with a line, and the resulting polygon was projected onto the basal plane for each time point. The polygon was color-coded according to its time point in the movie.

To calculate the cross-sectional area of the ring (*Figure 3C*), the centroid of the polygon was triangulated using its vertices. The area of each component triangle was calculated as half of the cross product of the two vectors formed by the centroid and the two adjacent vertices. The area of the ring was calculated as the sum of the areas of each component triangle. Ring areas were calculated for each movie time point.

To determine the apical-basal position of the E-cadherin::YFP ring, we calculated the orthogonal distance from the centroid of the ring to the basal plane. This distance was calculated as the dot product of two vectors: the unit normal vector of one of the basal measurement points, and a vector from the centroid to basal measurement point that was used as the origin of the unit normal vector.

### Analyses of the extruding nucleus

We defined the duration of nuclear extrusion (*Figure 3F*) as the length of time that the extruding cell's nucleus was moving apically. To determine this duration, Fiji-processed movies were opened from OME-TIFF files (*.ome.tif) in Bitplane Imaris and viewed as 3D volumes using the Surpass module. The nuclei of extruded enterocytes were digitally isolated via clipping planes and viewed in cross-section. The nuclei of enterocytes surrounding the extruding cell were used to establish the baseline level of the epithelium. The duration of nuclear extrusion was measured from the time point at which apical movement of the nucleus was first apparent to the time point of maximal apical displacement from the baseline. The centroid of the nucleus was calculated from surface-recognized objects in Imaris. The distance of the nucleus from the basal surface (*Figure 3D*) was calculated as the orthogonal distance from the centroid of the nucleus to the basal plane, as defined by the basal epithelium reference points described above.

## Spatiotemporal analyses of stem-cell mitoses

### Mitotic duration

Mitoses were identified by visual inspection of maximum intensity z-projections in Fiji and confirmed in Bitplane Imaris using the Surpass module for 3D visualization. To calculate the durations of individual mitoses (*Figure 3H*), we designated the start point as the initiation of nuclear condensation in the mother cell and the end point as the decondensation of the two sets of daughter chromosomes.

### Mitotic index

To calculate mitotic index, we used nine movies of *ubi-his2av::mRFP*-expressing midguts that each contained at least one identifiable division. Movies were processed in Fiji, and nuclei were identified and tracked in Imaris as described above. Mitotic index was calculated as $T_M/T_{SC}$, where $T_M$ is the sum of the durations of 39 individual mitoses in the 11 movies, and $T_{SC}$ is the sum of the durations of 'screen time' for all the stem cells in the same movies. Stem cell 'screen time' was calculated as the product of the number of stem cells at t=0 in a particular movie and the duration of that movie. (On occasion, stem cells disappeared or appeared over the course of a movie, but these events were infrequent and are not included in our calculations.) To determine the number of stem cells in a movie at t=0, we used one of two approaches. For midguts that expressed *esg >his2av::CFP* and *GBE-Su(H)-GFP:nls* in addition to *ubi-his2av::mRFP* (2 of 9 movies), stem cells were identified as CFP⁺ cells with GFP:RFP <0.17 (see*Figure 5*), and the number of stem cells was counted following Imaris surface recognition protocols as described above. For midguts in which stem cells were not identifiable through specific markers (7 of 9 movies), the number of stem cells was estimated as 20% of total *ubi-his2av::mRFP*-expressing cells (*de Navascués et al., 2012*; *O'Brien et al., 2011*).

### Horizontal-vertical orientation

To measure the real-time horizontal-vertical orientations of dividing cells (*Figure 4A–E*), Fiji-processed movies were opened as OME-TIFF files (*.ome.tif) in Bitplane Imaris. For each mitotic cell, the positions of the spindle poles and of the basal epithelial surface were determined at each time point between the start and the end of mitosis (*Figure 4—figure supplement 1*). Spindle pole positions were inferred from the morphology of the condensed chromosomes and marked using the Measurement Points tool. A plane representing the basal epithelium was defined using the Measurement Points tool to place three points on the basal epithelial surface underlying the spindle. To identify the position of the basal surface, we used either the basement membrane stain Conconavalin-A-Alexa647 or the background fluorescent signal of the cytoplasm of enterocytes made visible when movies were digitally overexposed.

The coordinates of the spindle poles and basal planes in 3D space were exported as Excel files and opened in Mathematica. To calculate the spindle angle, we defined two vectors: the 'spindle pole vector', which was the difference between the coordinates of the two spindle poles, and the 'basal plane vector', which was the cross product of two vectors determined from the three points defining the basal plane. The spindle angle was calculated as the dot product of the spindle pole vector and the basal plane vector.

### Longitudinal-circumferential orientation

To measure the longitudinal-circumferential orientation of dividing cells (*Figure 4F–H*), movies were analyzed as maximum-intensity projections in Fiji. Longitudinal and circumferential axes were determined for each mitotic cell by visual inspection, based on the local morphology of the midgut tube and the orientation of the ellipsoid nuclei of the surrounding enterocytes. Division orientation was measured at the time point when we observed decondensation of the daughter chromosomes, an event signifying the end of mitosis. To calculate longitudinal-horizontal orientation, we used the Fiji Angle Tool, which measures an angle defined by two vectors formed from three points. One vector was defined by the difference between the positions of the two daughter cells, and the other vector was defined by the longitudinal axis of the midgut tube.

### Orientation relative to neighboring enteroblasts

We identified mitoses in which the dividing cell contacted either one enteroblast or two enteroblasts using visual inspection. To determine the spatial coordinates of the mitotic cells and the

enteroblasts, Fiji-processed movies were opened as OME-TIFF files (*.ome.tif) in Bitplane Imaris. Nuclei were recognized using the Surface Recognition Wizard. The 3D coordinates of the relevant cells were exported into MATLAB.

Division orientations relative to neighboring enteroblasts (*Figure 4I–K*) were calculated as follows. For mitotic cells contacting one enteroblast, we designated the daughter cell closer to the enteroblast as 'D1' and the other daughter cell as 'D2'. We defined an 'enteroblast-D1 vector' as the difference between the coordinates of the enteroblast and D1. We defined a 'D1-D2 vector' as the difference between the coordinates of D1 and D2. To calculate the division angle, we computed the dot product of the enteroblast-D1 vector and the D1-D2 vector.

For mitotic cells contacting two enteroblasts, we designated the reference enteroblast as the enteroblast whose nucleus was closer to the mother stem cell nucleus prior to division. D1 and D2 daughters were determined relative to that enteroblast following the procedure detailed above.

Mitotic cells that contacted enteroblasts were excluded from analyses of horizontal-vertical and planar orientations.

## Quantitative assessment of Notch reporter activation

Activation of the Notch reporter *GBE-Su(H)-GFP:nls* was measured in movies of fate sensor midguts (*esgGal4, UAS-his2b::CFP, GBE-Su(H)-GFP:nls; ubi-his2av::mRFP*) (*Figures 5* and *6*). As described above, Fiji-processed movies were opened as OME-TIFF files (*.ome.tif) in Bitplane Imaris, and surface recognition was performed to identify individual cell nuclei using the Add New Surfaces function in the Surpass Module. To quantify *GBE-Su(H)-GFP:nls* activation, we calculated the normalized ratio of GFP:nls and His2av::mRFP intensities as follows. (1) To generate normalized intensity values, raw intensity values for GFP and RFP fluorescence of single cells at each individual time point were determined from cell nuclei. These raw intensities were exported to MATLAB and divided by the maximum intensity in that movie to yield normalized intensities. (2) The ratio of normalized GFP:RFP intensities was calculated for each cell at each time point. The resulting real-time, normalized GFP:RFPs enabled quantitative comparison of *GBE-Su(H)-GFP:nls* expression between different cells, at different times, or across different movies.

## Calculating inter-nuclear distances of progenitor pairs

To perform the initial analysis of inter-nuclear distances for contacting and non-contacting progenitor pairs (*Figure 6—figure supplement 1*), we used movies of midguts with genotype *esgGal4, UAS-LifeactGFP; ubi-His2av::RFP*. For this analysis, two *esg⁺* cells were designated as a pair if they were mutually closer to each other than to any other *esg⁺* cell. Pairs were selected randomly from single time points in four separate movies. Movies were examined in 4D using the Surpass Module in Imaris, and *esg⁺* pairs were identified as either contacting or non-contacting on the basis of their LifeactGFP signal. To determine the inter-nuclear distance of a pair, the (x,y,z) coordinates for the centroid of each nucleus were determined on the basis of surface recognition in Imaris. The distance D between the two centroids was calculated using the equation $D = \sqrt{(x_2-x_1)^2 + (y_2-y_1)^2 + (z_2-z_1)^2}$.

To determine the inter-nuclear distances for sibling pairs with known birth times (*Figure 6*), we used movies of 'fate sensor' midguts (*esgGal4, UAS-his2b::CFP, GBE-Su(H)-GFP:nls; ubi-his2av::mRFP*). Following a stem cell division, the inter-nuclear distance of the two siblings at each movie time point was calculated as described above.

## Acknowledgements

ENS was supported by NSF GRFP DGE-1656518 and by a Stanford Developmental Biology and Genetics NIH Training Grant (2T32GM00779038). PMR was supported by a Stanford Bio-X Bowes Graduate Fellowship and by a Stanford DARE (Diversifying Academia Recruiting Excellence) Fellowship. XD was supported by NRSA 1F32GM115065 and by a Stanford Dean's Postdoctoral Fellowship. LAKJ was supported by 1F31GM123736-01 and by a Stanford Developmental Biology and Genetics NIH Training Grant (2T32GM00779038). This work was supported by the National Institutes of Health (NIH) (R01GM116000-01A1), the Stanford Discovery Fund Innovation Program, and a Center for Biomedical Imaging at Stanford Seed Grant to LEO. Confocal microscopy was performed at the Stanford Beckman Cell Sciences Imaging Facility (NIH 1S10OD01058001A1). Fly extract was

obtained from the *Drosophila* Genomics Resource Center (NIH 2P40OD010949). We thank B Bolival for the construction of the fate sensor stocks; B Ohlstein, Y Inoue, D Montell, J de Navascués, and the Bloomington *Drosophila* Stock Center (NIH P40OD018537) for other *Drosophila* stocks; T Larsen and B Pruitt for assistance with and use of a micro laser cutter; M Goodman and T Blankenship for helpful discussions; and C Cabernard, A Bardin, J de Navascués, S Brantley and A Sherlekar for valuable comments on the manuscript.

## Additional information

### Funding

| Funder | Grant reference number | Author |
|---|---|---|
| National Institutes of Health | R01GM116000-01A1 | Judy Lisette Martin<br>Erin Nicole Sanders<br>Paola Moreno-Roman<br>Leslie Ann Jaramillo Koyama<br>Shruthi Balachandra<br>XinXin Du<br>Lucy Erin O'Brien |
| National Institutes of Health | Stanford Discovery Fund Innovation Program | Lucy Erin O'Brien |
| Stanford University | Center for Biomedical Imaging at Stanford Seed Grant | Judy Lisette Martin<br>Lucy Erin O'Brien |
| National Science Foundation | GRFP DGE-1656518 | Erin Nicole Sanders |
| National Institutes of Health | 2T32GM00779038 | Erin Nicole Sanders<br>Leslie Ann Jaramillo Koyama |
| William K. Bowes, Jr. Foundation | Stanford Bio X Bowes Graduate Fellowship | Paola Moreno-Roman |
| Stanford University | Stanford DARE (Diversifying Academia Recruiting Excellence) Fellowship | Paola Moreno-Roman |
| National Institutes of Health | NRSA 1F32GM115065 | XinXin Du |
| Stanford University | Stanford Dean's Postdoctoral Fellowship | XinXin Du |
| National Institutes of Health | 1F31GM123736-01 | Leslie Ann Jaramillo Koyama |

The funders had no role in study design, data collection and interpretation, or the decision to submit the work for publication.

### Author contributions

Judy Lisette Martin, Erin Nicole Sanders, Conceptualization, Resources, Data curation, Software, Formal analysis, Supervision, Funding acquisition, Validation, Investigation, Visualization, Methodology, Writing—original draft, Project administration, Writing—review and editing; Paola Moreno-Roman, Conceptualization, Resources, Data curation, Software, Formal analysis, Funding acquisition, Validation, Investigation, Visualization, Methodology, Writing—original draft, Writing—review and editing; Leslie Ann Jaramillo Koyama, Resources, Data curation, Formal analysis, Funding acquisition, Validation, Investigation, Visualization, Methodology, Writing—original draft, Writing—review and editing; Shruthi Balachandra, Resources, Data curation, Software, Formal analysis, Funding acquisition, Validation, Investigation, Visualization, Methodology, Writing—review and editing; XinXin Du, Resources, Software, Formal analysis, Funding acquisition, Validation, Methodology, Writing—original draft, Writing—review and editing; Lucy Erin O'Brien, Conceptualization, Formal analysis, Supervision, Funding acquisition, Validation, Visualization, Methodology, Writing—original draft, Project administration, Writing—review and editing

## Author ORCIDs
Lucy Erin O'Brien (iD) http://orcid.org/0000-0001-7660-2524

### Decision letter and Author response
Decision letter https://doi.org/10.7554/eLife.36248.045
Author response https://doi.org/10.7554/eLife.36248.046

## Additional files

### Supplementary files

• Source code 1. Registration macros utilizing the ImageJ plugin StackReg to perform three channel stack registration over time. In this macro, the XY negative space around the image is increased by a user-defined amount to account for the shifting of stack slices during the registration process. The movie is then collapsed into an RGB format and StackReg is performed on each time point using a loop function. Once completed, corrected time points are concatenated, converted back to three color hyperstacks, and then the ImageJ plugin Correct 3D Drift is applied to correct for global volume movement of the tissue over time. The macro is in *.ijm format which can be opened and viewed in ImageJ.
DOI: https://doi.org/10.7554/eLife.36248.039

• Transparent reporting form
DOI: https://doi.org/10.7554/eLife.36248.040

### Data availability

All data generated or analyzed during this study are included in the manuscript and supporting files. Source data files for figures have also been uploaded to Dryad (https://dx.doi.org/10.5061/dryad.1v1g1b0).

The following dataset was generated:

| Author(s) | Year | Dataset title | Dataset URL | Database and Identifier |
|---|---|---|---|---|
| Martin J, Sanders E, Moreno-Roman P, Jaramillo Koyama L, Balachandra S, Du X, O'Brien L | 2018 | Data from: Long-term live imaging of the Drosophila adult midgut reveals real-time dynamics of division, differentiation, and loss | https://dx.doi.org/10.5061/dryad.1v1g1b0 | Dryad Digital Repository, 10.5061/dryad.1v1g1b0 |

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
