## [Decision Letter]

Thank you for submitting your article "Long-term live imaging of the *Drosophila* adult midgut reveals real-time dynamics of division, differentiation, and loss" for consideration by *eLife*. Your article has been reviewed by Sean Morrison as the Senior Editor, a Reviewing Editor, and three reviewers. The reviewers have opted to remain anonymous.

The reviewers have discussed the reviews with one another and the Reviewing Editor has drafted this decision to help you prepare a revised submission.

Summary:

The paper details a new live-imaging method for the *Drosophila* intestine, and delivers several sets of very interesting quantitative data, nicely analyzed, that were collected using this method. The imaging method is a clear advance and has the potential to generate many new insights pertinent to intestinal epithelial homeostasis, if widely applied. Although technically it is not simple, the descriptions provided are sufficiently detailed, especially the excellent movie, to allow other researchers to apply it. The paper is well-written and nicely illustrated – excellent in overall quality. The data that are presented represent a new level of analysis (over standard work using fixed samples), for this interesting stem cell system.

Essential revisions:

The reviewers all recognize that a workable live-imaging platform for the *Drosophila* midgut represents a significant achievement. The paper only requires at least one example to show that novel insights can be obtained using this approach. The major concern is that none of the three examples put forward by the authors for this purpose were fully convincing due to the small sample sizes and some residual scientific concerns. The revised version of the paper should bring at least one of the examples to completion, by addressing the issues raised in the reviews.

Briefly, the point that spindles re-orient multiple times needs to address the concern that these observations might simply be the result of muscular waves going along the gut and not the sort of spindle re-orientation related to ISC biology. The delay in activating Notch signaling in EBs must address the concern that this is caused by the suboptimal conditions of the tissue during the live imaging procedure that is depressing the division rate and does not occur normally. The section on cell extrusion seems particularly promising, but currently several of the interesting observations are apparently based on an N=1. Additional data should be provided.

The revised version should also summarize the current limitations of the system and the prospects for further improvements.

*Reviewer #1:*

Martin et al. report the development of a system for live imaging of cells in the *Drosophila* midgut, a highly responsive tissue maintained by actively dividing intestinal stem cells. By cleverly employing a series of known techniques for genetically labeling, mounting, image-stabilizing, and 3-D reconstructing data collected from movies over many hours, a new appreciation of midgut cellular dynamics was obtained. This accomplishment is impressive, and the technology for live-imaging *Drosophila* midguts is very clearly described in the methods and in an excellent instructional movie.

The gut is a complex tissue that is known to interact with many if not most other tissues in the animal via enteroendocrine, neural peptide, neural and metabolic effectors. A major concern is the fact that the level of ISC divisions appears to be at least 5-fold reduced during live imaging, compared to their rate in vivo, with at least half of the interpretable movies lacking any cell divisions, despite continued viability and muscular activity. What currently limits imaging to 12-16 hours? This suggests that the picture of midgut activity reported may be more typical of an under-nourished or stressed tissue, than a fully normal one.

Despite this caveat, the authors report several interesting observations. Based on a sample size of 39 mitotic events, the ISC mitotic spindle's orientation is seen to re-orient once, or even twice during each division. The results reported seem fairly consistent with previous studies of ISC divisional orientation, with the new dynamic data being consistent with the passage of a wave of muscle contraction that might alter the orientation by changing the positions of larger nearby cells and generating a new mechanical equilibrium.

The activation of Notch signaling in newly formed enteroblasts (EBs) is described using a GBE-Su(H) reporter, and in the tissue at steady state. The authors describe a "delay" in the rate of activation of the reporter of 5-6 hours before levels consistent with the enteroblast state are observed. The physiological significance of such a delay is not well discussed by the authors, and the fact that only one daughter cell could be directly observed to EB fate heightened concern that cell production under the current conditions of live imaging are not optimal. The significance of a delay in Notch activation should be more thoroughly discussed. For example, would this result in over-production of Notch target genes that require low levels of receptor activation? The existence of the delay in free ranging animals and its physiological origin should be supported by fixed image data from normal and under-nourished animals from the same genetic background. It would be interesting to compare the delay to the kinetics of loss of Delta staining in these cells, as well.

Some of the most interesting results, although not stem cell related, were presented on the mechanics of cell extrusion, based on 34 studied examples. Single enterocytes are expelled by the coordinated action of six neighboring enterocytes in a coordinated process that propels their contents fully into the midgut lumen, and then cleaning seals the former region of contact. This section would benefit from experiments that provide some mechanistic insights into this fascinating process.

*Reviewer #2:*

In this manuscript, the authors present a novel protocol for live imaging that provides a valuable insight into the dynamics of division, differentiation and loss in the *Drosophila* midgut for up to 12-16 hours. Their approach significantly improves previous live imaging protocols and circumvents the caveats of attaining dynamic information from fixed samples as well as the live imaging ex vivo cultured samples that only last for up to 60-90 minutes. Their protocol therefore allows for imaging in a time-window that is long enough for better understanding real-time processes in vivo. The data from live imaging analysis provide support of the following:

– EC extrusion happens via the constriction of a pulsatile cadherin ring.

– Mitotic stem cells re-orient but if flanked between EBs, they are "anchored in place".

– Daughter cells delay Notch activation for many hours after birth.

In addition, the authors provide an automated analysis pipeline. Altogether, their new set-up reveals novel findings about the dynamics of tissue renewal and creates potential for many questions to be addressed in the future. Overall, the experiments are very well done, and this will be of broad interest.

The single EC extrusion analysis is based on one cell (n=1). The authors mention that 2 other cells have shown similar events but do not show any of these data. It will be important to provide description of at least a few other extrusions. Do they follow the same pulsatile contraction mode or are there many different ways for cell extrusion to occur?

The authors mention they see one example of the loss of a diploid cell but do not show whether this happens via the same mechanism? Could they please provide these data or comment on this?

They mention that cells can be extruded in groups of 2-5 cells but they do not comment on that further. Could the authors provide more information about this? Is it fundamentally different from the single cell extrusion?

An additional video of mitotic cell re-orientation would be helpful. Marianes and Spradling (2013) had described the oriented growth of clones as well as regional boundaries existing in certain subregions of the gut. Therefore, it is possible that the findings presented here regarding a lack of longitudinal-circumferential bias may be region-dependent. The authors might mention this possibility.

The authors carefully measured a ratio of GFP/RFP and associated this with active Notch signalling in enteroblasts. Do they expect that this ratio would need to be empirically determined for other labs with different microscopy set-ups and (presumably) if using different RFPs? If so, it would be helpful to explicitly state this somewhere.

Chen et al. (2018) have shown that there is another dividing cell type in the lineage that they term "Enteroendocrine Progenitor" cell (EEPs). Some of the dividing cells that were measure could be these precursors. This should be commented on.

*Reviewer #3:*

The paper details a new live-imaging method for the *Drosophila* intestine, and delivers several sets of very interesting quantitative data, nicely analyzed, that were collected using this method. The imaging method is a clear advance and has the potential to generate many new insights pertinent to intestinal epithelial homeostasis, if widely applied. Although technically it is not simple, the descriptions provide are sufficiently detailed to allow other researchers to apply it. The paper is well-written and nicely illustrated – excellent in overall quality. The data that is presented is likewise impressive in quality and represents a new level of analysis (over standard work using fixed samples), for this interesting stem cell system. The biological insights presented at this stage, while not surprising, do advance our understanding of the details of stem cell behavior and epithelial replacement in the fly gut. Overall, I feel this is a valuable unique work, and I support publication. A number of specific comments are listed in the next section – attending to these in review should enhance the usability of the method for other researchers.

1) It would be good to know the level of damage that the imaging setup induces in the fly. For example, does the dissection damage the heart? Is the occlusion of spiracles due to the agarose coating creating oxygenation issues and a hypoxic response in the fly? Are the trachea, being very fine and fragile structures, damaged during the dissection, especially considering that the area of dissection is close to major tracheal trunks? Is there any evidence of melanization or recruitment of adipocytes at the site of dissected cuticle? Please comment on these issues if possible.

2) Similarly, is prolonged imaging damaging the midgut epithelium? A staining with cell viability dyes on guts imaged for several hours could be helpful in this respect. High levels of DNA damage due to imaging may be a contributing factor to the lower than previously described mitotic index.

3) In the subsection “An apparatus for midgut imaging within live *Drosophila* adults”, it is stated that "Windows cut in the pliable ventral cuticle were unsuitable for long-term viability", although no explanation is given for this.

4) Towards the end of Video 2 the gut movements seem to drastically reduce in intensity. Also, many ECs seem to be extruded at this stage and progenitors increase in motility. Is this due to the animal dying? If so, what would be the best time window for imaging during which the animal is still mostly healthy?

5) In subsection “An apparatus for midgut imaging within live *Drosophila* adults”, it is stated that imaging at 29°C is not viable, however no explanation is given for the lower animal viability.

6) What is the percentage of movies blurred by animal/midgut movements? How many of said movies can be successfully corrected by software?

7) As for the automatic analysis, how reliably is cell identity assigned by software?

8) Is EC extrusion frequency the same between the first and last half of the movie?

9) In subsection “Enterocyte extrusion: Spatiotemporal dynamics of ring closure, ring travel, and nuclear travel”, it is stated that one instance of diploid cell extrusion was recorded. What was the cell type of said cell?

10) Instead of comparing the mitotic rate between the first and second half of the movie, it would be helpful analyze said parameter hourly, so that eventual stresses towards the end of the imaging session will not be averaged.

11) In the first two time-points of Figure 4C, the red bar seems to indicate the normal to the presumed orientation of the spindle, while in the 30' time-point it seems to be parallel to the direction the cells actually divided. It would therefore be useful to include also the intermediate time-point between the 15' and 30' frame.

12) Could the re-orientation during mitosis simply be due to artifacts due to peristalsis or the movement of neighboring cells? After all, there doesn't seem to be a pattern to these re-orientations discernible from Figure 4E.

13) Is circumferential oriented mitosis happening at a higher frequency (compared to a longitudinal orientation) near midgut region boundaries?

14) Are rearrangements of the mitotic axis behaving differently when happening near an enteroblast simply due to the greater motility of this cell type compared to enterocytes? (i.e. are ISC more spatially constricted when surrounded only by EC?)

15) If ISC division is influenced by the imaging procedure, how likely is differentiation affected as well in terms of the observed timing of N activation?

16) In Figure 6D we are shown two daughters from a cell whose mitosis was observed. It is quite interesting that upon contact, both cells have a spike in Su(H) expression, but it is the one with the smallest spike that will differentiate as described by its following steady increase in Su(H) expression. During contact, the cell which will maintain its stemness reaches a GFP:RFP ratio well above the threshold previously defined by the authors. Has this behavior been observed more than once during imaging? If so, this would indicate that perhaps some ISC may be mistaken for EB due to transient N activation. It would also be a good indication of bi-directional N signaling between ISC and EB.

17) GFP:RFP plots in Figure 5 show that activation of N signaling may be a slow process. As the authors have focused their attention on the interaction of sibling cells, they have admittedly not being able to follow such interactions to the fullest due to the time required to observe a cell divide prior to the formation of daughters. Could therefore a similar analysis of progenitor cell interactions be performed on neighboring cells (i.e. not necessarily siblings) present since the beginning of the movie? In this manner, interactions could be followed for longer time.

18) As imaging has been performed in young flies (i.e. still undergoing adaptive growth), results may not correctly describe midgut homeostasis for older (> 5 days old) flies. A few tests on more mature flies would be welcome.

---

## [Author Response]

Essential revisions:The reviewers all recognize that a workable live-imaging platform for the Drosophila midgut represents a significant achievement. The paper only requires at least one example to show that novel insights can be obtained using this approach. The major concern is that none of the three examples put forward by the authors for this purpose were fully convincing due to the small sample sizes and some residual scientific concerns. The revised version of the paper should bring at least one of the examples to completion, by addressing the issues raised in the reviews.Briefly, the point that spindles re-orient multiple times needs to address the concern that these observations might simply be the result of muscular waves going along the gut and not the sort of spindle re-orientation related to ISC biology. The delay in activating Notch signaling in EBs must address the concern that this is caused by the suboptimal conditions of the tissue during the live imaging procedure that is depressing the division rate and does not occur normally. The section on cell extrusion seems particularly promising, but currently several of the interesting observations are apparently based on an N =1. Additional data should be provided.

We have now examined two of these three phenomena, peristaltic contractions and enterocyte extrusions, in detail. Unfortunately, we were unable to assess the kinetics of Notch activation in intact, unperturbed animals due to technical obstacles (c.f. reviewer 1, fourth point; reviewer 2, eighth point).

For peristalsis, we found no temporal correlation between these and mitotic spindle reorientations; this analysis is described below in reviewer 3, twelfth point and is also now cited in the Results section. For extrusion, we extended our original analysis of a single extruding enterocyte to include two additional events; in all three events, we additionally examined instantaneous rates of ring constriction/relaxation (new Figure 3—figure supplement 1). These rate plots exhibit characteristic hallmarks of ratcheting, a finding that is novel and unexpected. Although ratcheting can drive cell deformations in embryonic morphogenesis, it has not been thought to drive cell extrusion in any context. Extrusion-associated ratchets show intriguing differences compared to cell-deforming ratchets in terms of time scale (minutes, not seconds) and subcellular localization (basal, not apical). Importantly, our 4D analyses provide, to our knowledge, the first time-resolved morphometric insights into this essential process for any tissue. We believe the expanded scope and biological interest of the extrusion data fulfill the criteria for completion as described by the Reviewing Editor.

The revised version should also summarize the current limitations of the system and the prospects for further improvements.

A summary of current limitations and prospects for further improvements is now included in the Conclusions.

Reviewer #1:Martin et al. report the development of a system for live imaging of cells in the Drosophila midgut, a highly responsive tissue maintained by actively dividing intestinal stem cells. By cleverly employing a series of known techniques for genetically labeling, mounting, image-stabilizing, and 3-D reconstructing data collected from movies over many hours, a new appreciation of midgut cellular dynamics was obtained. This accomplishment is impressive, and the technology for live-imaging Drosophila midguts is very clearly described in the Materials and methods section and in an excellent instructional movie.

We thank the reviewer for these assessments.

The gut is a complex tissue that is known to interact with many if not most other tissues in the animal via enteroendocrine, neural peptide, neural and metabolic effectors. A major concern is the fact that the level of ISC divisions appears to be at least 5-fold reduced during live imaging, compared to their rate in vivo, with at least half of the interpretable movies lacking any cell divisions, despite continued viability and muscular activity. What currently limits imaging to 12-16 hours? This suggests that the picture of midgut activity reported may be more typical of an under-nourished or stressed tissue, than a fully normal one.

We agree that the gut is not “fully normal”. Rather, we would argue that our platform is a major advance over ex vivo culture, which is the only other live imaging approach that has been reported to date. Long-term midgut imaging opens the door to many dynamic processes that have previously been unapproachable.

The reviewer cites two concerns. The first is why stem cell divisions are less frequent than in vivo, with some movies containing no identifiable divisions. We do not know the reason, but two likely contributors are: (1) The Schneider’s-based media that covers the dorsal window. Potential drawbacks are that (a) the media formulation may not be optimal; (b) the addition of media dilutes circulating factors present in the hemolymph; and/or (c) the media contains pen/strep, which is needed to prevent bacterial contamination but which also eliminates the commensal bacteria that support ‘normal’ rates of division (Buchon et al., 2009). (2) Phototoxicity. Most of the movies in our study used a confocal setup. With confocal, laser light illuminates the entire tissue volume. An imaging setup that exposes the sample to less laser light, such as multi-photon or spinning disk, might help to restore native division rates. We now discuss these issues in the Conclusion.

The second concern is why movie durations are limited to 12-16 hours. There are two major reasons: (1) Our shared microscope facility places an upper limit of 16 hours on most imaging sessions, so we have limited experience with movies longer than 16 hours. (2) After 12-16 hours, midguts often acquire one or more features that suggest diminished viability: wholesale dimming of fluorescent signals; enterocyte extrusions en masse (12 cells or more); loss of ordered enterocyte packing; appearance of multiple pyknotic nuclei; widespread flattening and spreading of progenitor cells. We believe that these events may be effects of the imaging media and/or phototoxicity, as described above. During post-acquisition processing, the appearance of any one of these criteria renders subsequent time points unsuitable for analysis. We now discuss these criteria in the Materials and methods section.

Despite this caveat, the authors report several interesting observations. Based on a sample size of 39 mitotic events, the ISC mitotic spindle's orientation is seen to re-orient once, or even twice during each division. The results reported seem fairly consistent with previous studies of ISC divisional orientation, with the new dynamic data being consistent with the passage of a wave of muscle contraction that might alter the orientation by changing the positions of larger nearby cells and generating a new mechanical equilibrium.

The idea that muscle contractions could trigger mitotic re-orientation is intriguing. However, for the 10 mitoses in Figure 4E, we found no temporal correlation between muscle contractions and re-orientations during the 15 minutes prior to each re-orientation event. This finding, which has now been added to the Results section indicates either that contractions are not the trigger for re-orientation, or that the effects of muscle contractility are exerted across periods longer than 15 minutes.

The activation of Notch signaling in newly formed enteroblasts (EBs) is described using a GBE-Su(H) reporter, and in the tissue at steady state. The authors describe a "delay" in the rate of activation of the reporter of 5-6 hours before levels consistent with the enteroblast state are observed. The physiological significance of such a delay is not well discussed by the authors, and the fact that only one daughter cell could be directly observed to EB fate heightened concern that cell production under the current conditions of live imaging are not optimal. The significance of a delay in Notch activation should be more thoroughly discussed. For example, would this result in over-production of Notch target genes that require low levels of receptor activation? The existence of the delay in free ranging animals and its physiological origin should be supported by fixed image data from normal and under-nourished animals from the same genetic background. It would be interesting to compare the delay to the kinetics of loss of Delta staining in these cells, as well.

The reviewer raises two main considerations:

1) Is the post-birth delay in Notch activation a native, physiological behavior that also occurs in free-ranging animals, or is it due to sub-optimal imaging conditions and/or nutritional stress? The question is valid. Unfortunately, it is also extremely difficult to address conclusively. Because midgut stem cells divide stochastically and asynchronously, birth times are unpredictable and, in intact animals, can only be estimated by techniques that label mitotic stem cells with a heritable marker. However, current techniques (e.g. FLP/FRT-based clonal labeling) rely on heat shock, which itself induces a stress response that affects division (Strand and Micchelli, 2011) and likely also differentiation (Buchon et al., 2010). To avoid this conundrum, we would need to develop new methodology that rapidly and heritably marks midgut cells at birth without causing stress or other perturbation. This endeavor would require substantial time and effort and have no guarantee of success. Indeed, the technical obstacles of fixed-gut approaches highlight the particular strength of live imaging to reveal dynamic new phenomena. Because of these challenges, we cannot exclude that the imaging protocol itself contributes to the delay. However, we note that kinetics of Notch activation are similar irrespective of whether cells activate Notch at the beginning or end of an imaging session. This similarity suggests that the delay is not due to cumulative stresses of imaging. To address the reviewer’s comment, we now discuss these issues in the text (subsection “Real-time kinetics of enteroblast transitions”; subsection “A delay in Notch activation”).

2) The reviewer points out that potential biological implications of delayed/slow Notch activation were not discussed. We have now included this discussion in the text. The implications fall into two categories: (a) The post-birth delay in initiation of Notch-activated transcription (i.e. when does activation start). This is now discussed in subsection “A delay in Notch activation”. (b) The slow rate at which levels of Notch activation, once initiated, increase to the enteroblast threshold (i.e. how quickly does activation increase). This is now discussed subsection “Contacts between newborn siblings are variable and dynamic”. The main points of these two discussions, together with some ancillary comments, are below:

a) Implications of post-birth delay. We speculate that this delay may facilitate the particular ability of midgut stem cells to undergo population-level shifts in symmetric vs asymmetric fate outcomes (O’Brien, Jin Jiang, Jasper, Sokol). A built-in ‘waiting period’ could allow newborn cells to receive and integrate a large variety of intrinsic and extrinsic signals before deciding to differentiate. How such a delay is set in place at molecular and cellular levels—i.e. whether Notch is kept OFF through regulatory events at the plasma membrane, in endosomes, or in the nucleus—will be an important topic for future work.

b) Implications of slow rate of Notch activation. Slow activation could signify that a large number of Delta interactions must accumulate before a cell’s level of activated Notch becomes sufficient to reach the enteroblast threshold. Such a stringent requirement might ensure that enteroblasts are generated only when they are truly needed. A second, intriguing notion is that because Notch activation is slow, it can more easily be reversed. In our movies, we observed that nascent enteroblasts occasionally lose expression of *GBE-Su(H)-GFP:nls* and revert to stem-like status (subsection “Real-time kinetics of enteroblast transitions”). By lengthening the time between when cells become specified as enteroblasts and when they become irreversibly committed, slow activation could provide more chances to ‘backtrack’ if the tissue environment changes.

In addition, the reviewer wonders whether slow activation results in prolonged and predominant expression of high-affinity Notch targets. We think this scenario is likely. Perhaps a larger issue is whether the high-affinity targets are also the most important. What is/are the essential target(s) for NICD/CSL in nascent enteroblasts, and what is/are the nature of their CSL binding sites? Bardin et al. (2010) showed that these targets include the E(spl)-C complex. However, not all E(spl) genes are required (Lu et al., 2015); those that are required have not yet been identified; and other, unknown Notch targets are also needed (Bardin et al., 2010; Chen et al., 2018). In general, Notch targets exhibit two types of activation kinetics: graded, dose-dependent expression, or binary, ON/OFF expression. The GBE-Su(H) reporter in our movies, with its synthetic, high-affinity CSL sites (Furriols and Bray, 2001; de Navascues et al., 2012), likely has graded, dose-dependent expression. However, the fact that a sharp threshold of *GBE-Su(H)-GFP:nls* distinguishes stem cells and enteroblasts (Figure 5C) may hint that one or more essential targets are under binary, ON/OFF control.

Some of the most interesting results, although not stem cell related, were presented on the mechanics of cell extrusion, based on 34 studied examples. Single enterocytes are expelled by the coordinated action of six neighboring enterocytes in a coordinated process that propels their contents fully into the midgut lumen, and then cleaning seals the former region of contact. This section would benefit from experiments that provide some mechanistic insights into this fascinating process.

We appreciate the reviewer’s enthusiasm for the findings related to homeostatic cell extrusion. In the revised manuscript, the extrusion studies have been expanded to include (1) a new figure supplement (Figure 3—figure supplement 1), with 6 panels of new data analyses, and (2) a new video (Video 10). These data are discussed in the Results section and summarized below:

New Figure 3—figure supplement 1 provides strong evidence that extrusion of enterocytes occurs via a ratcheting mechanism. In the original manuscript, this notion was based on the detailed measurements of closure kinetics for one extrusion (Figure 3A-C). In the revision, we have extended this analysis to two additional, independent extrusions (Figure 3—figure supplement 1B-C). In addition, we have performed new analyses of instantaneous rates of ring closure over time for all three extrusions (Figure 3—figure supplement 1A’-C’). The dynamics of the three ring closures are strikingly similar, consistent with a shared regulatory mechanism. Importantly, recurrent switching between negative rates (constriction) and near-zero/positive rates (stabilization/relaxation) is a hallmark of ratcheting (e.g. Martin et al., 2009; Rauzi et al., 2010). New Video 10 shows an extrusion of an enteroendocrine cell. Enteroendocrine extrusions are ~10x less frequent than enterocyte extrusions, so quantitative analyses were performed solely on enterocytes.

These analyses provide, to our knowledge, the first time-resolved morphometric insights into cell extrusion for any tissue in vivo. In particular, our finding that extrusion involves ratcheted constrictions is novel and possibly revealing. Although ratcheting is known to drive cell deformation in embryonic morphogenesis, it has not been thought to drive cell extrusion in any context. Extrusion-associated ratchets exhibit intriguing differences from developmental ratcheting in terms of time scale (minutes, not seconds) and subcellular localization (basal, not apical). Elucidating the molecular regulation of homeostatic cell extrusion and ratcheting is of high interest—but beyond the scope of this study, whose purpose is to report a new technical approach. We are investigating this molecular regulation and hope to report these findings in a future paper.

Reviewer #2:[…]The single EC extrusion analysis is based on one cell (n=1). The authors mention that 2 other cells have shown similar events but do not show any of these data. It will be important to provide description of at least a few other extrusions. Do they follow the same pulsatile contraction mode or are there many different ways for cell extrusion to occur?

We appreciate the reviewer’s point and have extended our original analysis to include the two additional extrusions, for n=3 total (Figure 3—figure supplement 1). In addition, for all three extrusions we have added new plots that show instantaneous rates of ring constriction/relaxation over time. The frequency switching between negative rates (constriction) and near-zero/positive rates (stabilization/relaxation) is a hallmark of ratcheting (e.g. Martin et al., 2009; Rauzi et al., 2010) and is exhibited by all three extrusions. Indeed, the overall dynamics of the three ring closures are strikingly similar in terms of pulse duration, rates of constriction/stabilization, and total time for closure to complete. These similarities, which are detailed in the revised Results section, are consistent with a common regulatory mechanism.

The authors mention they see one example of the loss of a diploid cell but do not show whether this happens via the same mechanism? Could they please provide these data or comment on this?Yes, new Video 10 shows the extrusion of this diploid cell. We believe the cell is enteroendocrine since it is esg^—^, Su(H)^—^.They mention that cells can be extruded in groups of 2-5 cells but they do not comment on that further. Could the authors provide more information about this? Is it fundamentally different from the single cell extrusion?

We are currently exploring these ‘cluster’ extrusions more closely. Two speculations are that (1) they are triggered by peristaltic contractions, and/or (2) they are a specific response to tissue crowding.

An additional video of mitotic cell re-orientation would be helpful.

An additional video of mitotic re-orientation is now included (new Video 13).

Marianes and Spradling (2013) had described the oriented growth of clones as well as regional boundaries existing in certain subregions of the gut. Therefore, it is possible that the findings presented here regarding a lack of longitudinal-circumferential bias may be region-dependent. The authors might mention this possibility.

We thank the reviewer for raising this interesting possibility. In our analysis, all mitoses occurred in the midgut’s R4a-b (P1-2) region, which is the region exposed by the cuticular window. Whether or not circumferential-longitudinal orientation is similar in other regions, or at region boundaries, is an open question. This point is now mentioned in the Results section.

G) The authors carefully measured a ratio of GFP/RFP and associated this with active Notch signalling in enteroblasts. Do they expect that this ratio would need to be empirically determined for other labs with different microscopy set-ups and (presumably) if using different RFPs? If so, it would be helpful to explicitly state this somewhere.

This point is valid; the precise numerical value of the enteroblast threshold

(GFP:RFP=0.17) is likely specific to our particular equipment and imaging parameters. We now mention this issue and the need for other labs to empirically re-assess the threshold (subsection “A quantitative threshold of Notch activation distinguishes stem cells and enteroblasts”).

Chen et al. (2018) have shown that there is another dividing cell type in the lineage that they term "Enteroendocrine Progenitor" cell (EEPs). Some of the dividing cells that were measure could be these precursors. This should be commented on.

We appreciate the reviewer’s comment and now discuss the possibility of EEP divisions in the text (subsection “Stem cell division: Mitotic orientation in real time”).

Reviewer #3:[…]1) It would be good to know the level of damage that the imaging setup induces in the fly. For example, does the dissection damage the heart? Is the occlusion of spiracles due to the agarose coating creating oxygenation issues and a hypoxic response in the fly? Are the trachea, being very fine and fragile structures, damaged during the dissection, especially considering that the area of dissection is close to major tracheal trunks? Is there any evidence of melanization or recruitment of adipocytes at the site of dissected cuticle? Please comment on these issues if possible.

Opening the cuticle would reasonably be expected to have some damaging effects. Manual dexterity is an important factor in minimizing this physical damage.

When performed as illustrated in the tutorial (Video 1), the heart is not physically touched and the left thoracic spiracles remain unoccluded.

We attempted to assess hypoxia using the ODD:GFP reporter (Kim et al., 2018). However, the results were ambiguous. We did not observe expression of ODD:GFP in midguts under our imaging conditions, but we also did not observe expression in positive control midguts subjected to hypoxia.

The major tracheal trunks and most secondary/tertiary branches remain intact, although occasional severing of smaller branches can occur (e.g. Video 3). To show the typical integrity of the tracheal network, we have added new Video 2, which shows trachea marked with *breathless>GFP*. The smaller tracheal branches move in synchrony with peristaltic contractions of the midgut, which indicates that the trachea and the midgut remain intimately associated during imaging. A large branch that is continuous with the smaller branches is visible at the upper right. The large branch does not move during peristalsis because it is not physically associated with the midgut; rather, it is a connector between the midgut-associated branches and a spiracle.

We have not observed melanization or recruitment of adipocytes.

2) Similarly, is prolonged imaging damaging the midgut epithelium? A staining with cell viability dyes on guts imaged for several hours could be helpful in this respect. High levels of DNA damage due to imaging may be a contributing factor to the lower than previously described mitotic index.

To gauge midgut cell viability during imaging, we added the cell death stain Sytox Green (Liang et al., 2017; Kolahgar et al., 2015) to the media prior to imaging. We then monitored the number of Sytox^+^ cells over time. In 3 extended movies, we observed that 2-7% of total cells became Sytox^+^ over the course of imaging. These data are in new Figure 1—figure supplement 3 and Video 5 and are discussed in the Results section (subsection “An apparatus for midgut imaging within live *Drosophila* adults”).

We expect that the midgut epithelium, like all living tissue, incurs photodamage as a consequence of fluorescence illumination, and it would not be surprising if photodamage contributes to lowered mitotic index. The probability of a mitotic event does not decrease with increased imaging times. Two possible interpretations of this finding are that: (1) pre-mitotic cells are hypersensitive to laser light, so short and long exposures are similarly inhibitory, or alternately that (2) lowered mitotic index is primarily caused by another factor(s).

3) In the subsection “An apparatus for midgut imaging within live Drosophila adults”, it is stated that "Windows cut in the pliable ventral cuticle were unsuitable for long-term viability", although no explanation is given for this.

A ventral window was unsuitable for two reasons: (1) Our imaging mount is incompatible with ventrally mounted animals. Specifically, the abdominal area-to-be-imaged must protrude through the cutout in the metal shim and be immersed in media; at the same time, the legs, spiracles, proboscis, and feeder tube must be on the opposite side of the shim, sealed off from the media-containing compartment. When the animal is mounted ventrally, the legs and spiracles cannot be sealed off because they are also on the ventral body. In addition, the feeder tube, which is essential for viability, cannot be properly positioned because the proboscis extends ventrally. (2) The relative softness of the ventral cuticle. Whereas the stiffness of the dorsal cuticle makes it easier to cut a window with clean, precise edges, the pliable nature of the ventral cuticle causes it to stretch and deform as the window is cut, resulting in windows with ragged edges and unpredictable dimensions. We have added a paragraph explaining these points to the Methods (subsection “Animal Preparation”).

4) Towards the end of Video 2 the gut movements seem to drastically reduce in intensity. Also, many ECs seem to be extruded at this stage and progenitors increase in motility. Is this due to the animal dying? If so, what would be the best time window for imaging during which the animal is still mostly healthy?

The reviewer is correct that, the end of Video 3 (originally Video 2) contains an increased frequency of enterocyte extrusions and progenitor motility. This change is particularly evident in the region of the gut tube at the bottom of the frame of view. The reviewer is also correct that the increased frequency of these behaviors is an indicator of tissue stress and decreased viability. In this case, however, viability was lost not because the animal died (the animal was still alive at the end of the imaging session). Rather, we believe that viability was lost because the midgut tube ruptured late in the imaging session. The site of the rupture, which was out of frame (but identifiable in post-imaging dissection), was close to these stress-associated cell behaviors. The reviewer asks about the best time window for acquiring healthy movies. We gauge midgut health using morphological criteria, rather than absolute time, because midgut health varies due to factors such as the exact placement of the cuticular window, animal-to-animal differences in midgut looping, and other, unknown factors. As detailed in the Methods (subsection “Movie registration and cell masking in Fiji”), we consider viability to be compromised when any one of the following behaviors are observed: (1) wholesale dimming of fluorescent signals; (2) enterocyte extrusions en masse (>12 cells at once); (3) loss of ordered enterocyte packing; (4) appearance of multiple pyknotic nuclei; (5) widespread flattening and spreading of progenitor cells. Following appearance of any of these criteria, subsequent timepoints were not used for analysis.

5) In subsection “An apparatus for midgut imaging within live Drosophila adults”, it is stated that imaging at 29°C is not viable, however no explanation is given for the lower animal viability.

We made ~5 attempts to image at 29°C; in all these attempts, the midgut tube ruptured within the first 3 hours of imaging. We believe these ruptures were a consequence of the animals’ heat-induced hyperactivity. Compared to animals at room temperature, animals at 29°C exhibited frequent, vigorous movements, including pronounced extension/compression of their abdomens. We think that these movements caused the midgut tube to be abraded against the sharp edges of the cuticular window, resulting in tube rupture. A brief explanation has now been added to the Results section.

6) What is the percentage of movies blurred by animal/midgut movements? How many of said movies can be successfully corrected by software?

Approximately 90% of movies exhibited blurring due to movements of the whole animal or of the midgut itself. Of these, ~70% were successfully corrected using the algorithms described in the Methods (subsection “Movie registration and cell masking in Fiji”). These percentages are now cited in the Results section.

7) As for the automatic analysis, how reliably is cell identity assigned by software?

With His2av::RFP as a cell marker, Imaris reliably identifies and tracks 70-90% of individual cells. This percentage is now cited in the Materials and methods section (subsection “Cell identification and tracking in Imaris”). Tissue movement is the major reason that some movies have lower rates of accurate identification.

8) Is EC extrusion frequency the same between the first and last half of the movie?

Yes, extrusion frequencies are comparable. We examined the timing of 17 enterocyte extrusions in 5 independent movies, each at least 8 hours in length. (One extrusion in Figure 3F occurred in a movie shorter than 8 hours; this movie was not included in this analysis.) Of the 17 extrusions in these 5 movies, 7 occurred in hours 0-4 and 10 occurred in hours 4-8. This result is now stated in the text (subsection “Enterocyte extrusion: Spatiotemporal dynamics of ring closure, ring travel, and nuclear travel”).

9) In subsection “Enterocyte extrusion: Spatiotemporal dynamics of ring closure, ring travel, and nuclear travel”, it is stated that one instance of diploid cell extrusion was recorded. What was the cell type of said cell?

The diploid cell was enteroendocrine (negative for expression of *esg>his2b::CFP*). A video of this extrusion has now been added (new Video 10).

10) Instead of comparing the mitotic rate between the first and second half of the movie, it would be helpful analyze said parameter hourly, so that eventual stresses towards the end of the imaging session will not be averaged.

As requested, we have now calculated the hourly mitotic index for each of imaging hours 1-14 (Author response image 1). These calculations used 35 mitoses that occurred in 8 independent movies. Similar to the formula for overall mitotic index (Materials and methods, subsection “Spatiotemporal analyses of stem cell mitoses”), the hourly mitotic index was calculated as T_M(h)_/T_SC(h)_, where h is the hour of imaging (h=0 to 14), T_M(h)_ is the sum of the durations of all individual mitoses that occurred during hour h for each of the 8 movies, and T_SC(h)_ is the sum of’ screen time’ of all individual stem cells during each hour of imaging for the same 8 movies. The number of individual stem cells in each movie was set as the number of individual stem cells at t=0; this number was determined as previously described in the Materials and methods (aforementioned subsection).

**Author response image 1. respfig1:** Hourly mitotic indices. Individual points show the mitotic index for hours 1-14 of live imaging. Earlier hours are denoted by lighter colors, and later hours are denoted by darker colors. Box shows the mean of the hourly indices (bar) and the first and third quartiles (0.18% and 0.38%). Whiskers show the standard deviation (0.19%). No correlation between mitotic index and imaging hour is observed.

As expected, the average of the 14 hourly mitotic indices (0.29% +/- 0.19%) was near-identical to the overall mitotic index reported in the original submission (0.28%). Half of the hourly mitotic indices (7/14 indices) clustered within a range of 0.21-0.31%. The other half fell outside this range, with individual values varying from 0.00% to 0.71%. We found no drift in mitotic index over time nor any discernible pattern in the timing of mitotic events. This finding suggests that mitotic rate is not principally determined by the cumulative stresses of imaging, but rather by factors that are already present at the start of imaging and/or that happen stochastically.

11) In the first two time-points of Figure 4C, the red bar seems to indicate the normal to the presumed orientation of the spindle, while in the 30' time-point it seems to be parallel to the direction the cells actually divided. It would therefore be useful to include also the intermediate time-point between the 15' and 30' frame.

We apologize for the ambiguity. In all three panels of Figure 4D (originally Figure 4C), the red bar is parallel to the presumed orientation of the spindle. To clarify, we have added a cartoon schematic of the mitotic cell below each panel.

12) Could the re-orientation during mitosis simply be due to artifacts due to peristalsis or the movement of neighboring cells? After all, there doesn't seem to be a pattern to these re-orientations discernible from Figure 4E.

The cause(s) of mitotic re-orientations is an intriguing topic. Peristalsis does not appear to be the immediate trigger, as we found no temporal correlation between peristaltic contraction and re-orientation for the 10 mitoses in Figure 4E. (This finding is now discussed in subsection “Stem cell division: Mitotic orientation in real time”.) Neighbor cell movements are another potential contributor. We will quantitatively analyze neighbor movements, as well as other stem cell-extrinsic and -intrinsic factors, in future work.

13) Is circumferential oriented mitosis happening at a higher frequency (compared to a longitudinal orientation) near midgut region boundaries?

The question is pertinent. As shown by Marianes and Spradling (2013), stem cell clones do not cross most region boundaries; one possible mechanism could involve circumferential orientation of boundary-localized divisions. Unfortunately, our current movies do not allow us to assess this interesting possibility. Because of animal-to-animal variation in midgut looping and placement of the cuticular window, the P1-2 boundary is not consistently exposed. Even when it is exposed, this particular boundary is hard to pinpoint because it lacks a pronounced muscle constriction. It would be precarious to try to identify the boundary-localized divisions, which are just a small fraction of all divisions, without live, region-specific markers. Rather, we have now added discussion of the possibility that boundary-localized divisions could be a special, circumferentially biased type of division (subsection “Stem cell division: Mitotic orientation in real time”).

14) Are rearrangements of the mitotic axis behaving differently when happening near an enteroblast simply due to the greater motility of this cell type compared to enterocytes? (i.e. are ISC more spatially constricted when surrounded only by EC?)

The possibility that neighbor cells somehow influence or even control mitotic reorientations is highly intriguing. Clearly, rearrangements of the mitotic axis are different when a stem cell has two neighboring enteroblasts (Figure 4I-K). The two enteroblasts serve to constrain the stem cell spindle to the enteroblast-enteroblast axis. We don’t know the mechanism for this stabilization, but we speculate that it may involve engagement of cell-cell adhesion receptors at stem cell-enteroblast interfaces and linkage of these receptors’ intracellular domains to astral microtubules.

Mitotic stem cells with exactly one enteroblast do not appear stabilized, perhaps because the spindle in this arrangement can pivot around its sole enteroblast ‘anchor’. We have not examined whether re-orientations are different when stem cells have one neighbor enteroblast compared to when they have neighbor enterocytes exclusively, or whether any such differences are due to enteroblast motility or spatial packing.

15) If ISC division is influenced by the imaging procedure, how likely is differentiation affected as well in terms of the observed timing of N activation?

The question is a valid one. We now discuss the possibility of such imaging-associated effects in the Results (subsection “Real-time kinetics of enteroblast transitions”; subsection “A delay in Notch activation?”). Although these effects cannot be excluded, we note that kinetics of Notch activation are similar irrespective of whether cells activate Notch at the beginning or end of an imaging session. This similarity suggests that any such effects would be due not to cumulative stresses of imaging, but rather to factors that were already present at the start of imaging and/or that occurred stochastically.

16) In Figure 6D we are shown two daughters from a cell whose mitosis was observed. It is quite interesting that upon contact, both cells have a spike in Su(H) expression, but it is the one with the smallest spike that will differentiate as described by its following steady increase in Su(H) expression. During contact, the cell which will maintain its stemness reaches a GFP:RFP ratio well above the threshold previously defined by the authors. Has this behavior been observed more than once during imaging? If so, this would indicate that perhaps some ISC may be mistaken for EB due to transient N activation. It would also be a good indication of bi-directional N signaling between ISC and EB.

We apologize that our description of the data was unclear. In Figure 6D, the spikes of GFP:RFP from t=2.6-3.5 h (Cell 1) and t=1.5-3.6 h (Cell 2) are artifacts. These artifacts were caused when a third, non-sibling cell collided with the two siblings (Video 18). This third cell was a mature enteroblast with an extremely bright GBE-Su(H)-GFP:nls signal. During the collision, when the cells were in intimate proximity, the bright GFP of the mature enteroblast ‘bled into’ the Imaris-recognized surfaces of the two siblings. The points in the Figure 6D graph show all GFP:RFP measurements, including those that were artifactually elevated by the collision. To better distinguish which measurements are artifactual, we have now made two modifications: (1) Gray, instead of black, points denote GFP:RFP measurements during the collision. (2) The interpolated lines now contain gaps when the collisions occur. In addition, we have added explanatory text to the Results section (subsection “Contacts between newborn siblings are variable and dynamic”) and improved the Figure 6D legend to describe this artifact more clearly.

Although the spikes in Figure 6D were artifactual, we did observe other instances of spiking that were endogenous. Figure 6B is one such example. The values of these spikes were consistently below the 0.17 enteroblast threshold, so they would not have caused stem cells to be misidentified as enteroblasts. We did not observe temporally correlated spikes between sibling cells; however, GBE-Su(H) may not be optimal for reporting possible bi-directional signaling.

17) GFP:RFP plots in Figure 5 show that activation of N signaling may be a slow process. As the authors have focused their attention on the interaction of sibling cells, they have admittedly not being able to follow such interactions to the fullest due to the time required to observe a cell divide prior to the formation of daughters. Could therefore a similar analysis of progenitor cell interactions be performed on neighboring cells (i.e. not necessarily siblings) present since the beginning of the movie? In this manner, interactions could be followed for longer time.

We thank the reviewer for this suggestion. Using the criteria for cell-cell contact from Figure 6—figure supplement 1A-B, we have now evaluated the four enteroblast-transitioning cells in Figure 5D-E. Strikingly, all four of these cells are in likely contact with stem cells immediately before they start to transition. Further, these contacts persist during much of the cells’ transition periods (gray background shading in graphs in Figure 5D-E). This correlation is consistent with the notion that prolonged contact with a stem cell—possibly, but not necessarily, the mother stem cell—may be required and/or sufficient for cells to transition to an enteroblast state. Explicit investigation of this scenario will be an attractive focus for future work.

18) As imaging has been performed in young flies (i.e. still undergoing adaptive growth), results may not correctly describe midgut homeostasis for older (> 5 days old) flies. A few tests on more mature flies would be welcome.

We are excited that our imaging methodology enables the comparison of real-time cell dynamics during homeostasis and growth. Such work is currently in progress, and we plan on reporting these results in a separate study.